# Real-time and high precision feature matching between blur aerial images

Dongchen Dai[1,2], Lina Zheng[1]*, Guoqin Yuan[1], He Zhang[1], Yu Zhang[1,2], Haijiang Wang[1,2], Qi Kang[1]

1 Changchun Institute of Optics, Fine Mechanics and Physics, Chinese Academy of Sciences, Changchun, Jilin, China, 2 University of Chinese Academy of Sciences, Beijing, China

* ailsazheng@163.com

**Data Availability Statement:** All relevant data are within the paper and its Supporting information files.

**Funding:** The authors received no specific funding for this work.

## Abstract

When aerial cameras get aerial remote sensing images, the defocus will occur because of reasons such as air pressure, temperature and ground elevation changes, resulting in different image sharpness of continual aerial remote sensing images. Nowadays, the rapidly developing feature matching algorithm will rapidly reduce the registration rate between images with different image sharpness. Therefore, in order to enable aerial cameras to get image sharpness parameters according to the locations of aerial image feature points with inconsistent sharpness, this paper proposes a feature matching algorithm between aerial images with different sharpness by using DEM data and multiple constraints. In this paper, the feature matching range is extended according to the modified aerial imaging model and the nonlinear soft margin support vector machine. Then the relative moving speed and its variation of the feature points in the image are obtained by using the extended L-k optical flow, and finally the epipolar geometric constraint is introduced. To locate the feature points is obtained under multiple constraints, there is no need to calculate the feature point descriptors, and some mismatched point pairs are corrected, which improves the matching efficiency and precision. The experimental results show the feature matching precision of this algorithm is more than 90%, and the running time and matching precision can meet various application needs of aerial cameras.

## Introduction

Aerial camera has become the main means to get geographic information because of its flexibility and timeliness, and it is commonly used in military reconnaissance, topographic mapping, disaster early warning and other fields [1–4]. According to its different working mode, aerial camera can be subdivided into push-broom type, swing-sweep type and frame-spoke type. When the swing-sweep type aerial camera gets the aerial remote sensing image, oblique camera swings perpendicular to the flight direction to get the ground information during the forward flight of the aircraft, to expand the field of view [5]. In the process of getting aerial remote sensing images, aerial cameras will produce defocus because of the influence of temperature, pressure and ground elevation difference, resulting in different sharpness of

**Competing interests:** The authors have declared that no competing interests exist.

continual aerial remote sensing images [6, 7]. Commonly used aerial camera image sharpness detection methods include photoelectric auto-collimation method, program control method and image processing method. Because the image processing method can use the computer to complete the task of image sharpness detection of aerial camera in real-time, and the mechanical structure is simple, it has become the principal method for the use and research of aerial camera [8–11]. However, the image processing method is influenced by the proportion of high frequency information in aerial remote sensing images. In the area where the ground scenery is rich, aerial remote sensing images got by aerial cameras are rich in high frequency information. Therefore, the method based on image processing is usually used to detect the clarity of aerial images, and adjust the mechanical structure to get clear aerial remote sensing images. However, in the areas such as oceans and deserts, which occupy more than half of the earth, there are few ground scenery and less high frequency information of aerial remote sensing images, so the image processing method can't be used to detect the image sharpness. To meet the demand of real-time sharpness detection of aerial camera in weak feature area, the method of feature matching offset is used to get sharpness detection parameters. Its important premise is to realize the feature matching algorithm in two aerial remote sensing images with different sharpness.

Image registration is an essential precondition for image processing, especially for image mosaic, 3D reconstruction and object tracking [12]. Image registration is commonly used in medical images, remote sensing images, video surveillance, computer vision and other fields. The purpose of remote sensing image registration is to geographically align two or more images containing overlapping scenes. Remote sensing image registration methods can be subdivided into feature-based methods and region-based methods. Among them, the region-based method realizes the correspondence by evaluating the similarity of the window pairs in the two images. In spatial domain, the commonly used similarity measures are Sum of Square Differences (SSD), Normalized Cross-Correlation (NCC) and Mutual Information (MI). The calculation of the Sum of Square Differences (SSD) is easy and fast, but it is sensitive to noise and intensity differences [13]. Normalized Cross-Correlation (NCC) is robust to linear strength changes, but it can't adapt to complex strength changes [14]. Mutual Information (MI) can solve the nonlinear intensity difference, but it ignores the spatial information of adjacent pixels, reduces the quality of image registration, and its computational cost is high [15]. In the frequency domain, the commonly used similarity measure is Phase Correlation (PC). Phase Correlation (PC) uses image intensity information and spatial similarity measurement to evaluate similarity. It has high computational efficiency, but can't deal with significant intensity differences [16]. Y. X. Ye proposes a multimodal image local descriptor: Chanel Feature of Orientated Gradient (CFOG), which describes the image pixel by pixel, and uses Fast Fourier Transform (FFT) to define a fast feature-based similarity measure. Chanel Feature of Orientated Gradient (CFOG) has the characteristics of high computational efficiency, but it can't deal with large differences in the rotation and scale [17]. H. M. Mohammed introduces the geometric and radiation characteristics of image pairs, and combines the region-based matching method to achieve uniformly distributed feature matching with more numbers and higher precision [18]. The region-based method calculates the similarity measure of image intensity, which evaluates the similarity measurement of all positions in the image region and search window by brute force search method, which can achieve high precision, but its running time is long.

The feature-based method is to determine the corresponding relationship by matching the local features between the images. The main local features include point features, contour features, edge features and regional features. Because local invariant features are robust to geometric and illumination differences, most image registration algorithms are based on local

invariant features. In general, the image registration method based on local features mainly includes three steps: 1. feature extraction; 2. feature description; 3. feature matching. The most popular local descriptor based on distribution is Scale Invariant Feature Transform (SIFT) proposed by Lowe. SIFT has strong discrimination capacity, and it is robust to scale change, rotation, illumination and so on, but its computational complexity is high [19, 20]. Gradient Location-Orientation Histogram (GLOH) proposed by K. Mikolajczyk, it uses the logarithmic polar coordinate position grid of 17 location elements instead of the 4 * 4 grid defined in the SIFT as the descriptor structure, but its computational cost is too high [21]. Partial Intensity Invariant Feature Descriptor (PIIFD) proposed by J. Chen uses symmetrical gradient direction histogram for remote sensing image registration. It uses the position and direction of edges in the spatial structure of SIFT to generate scalable binary edge images [22]. Speeded Up Robust Features (SURF) proposed by H. Bay is the accelerated version of SIFT, which uses Haar wavelet response, but its running speed still lags behind the current application needs [23]. DAISY proposed by E. Tola is a fast computing local feature descriptor for dense feature extraction. It uses Gaussian convolution to block aggregation the gradient direction histogram and quickly extract the feature descriptor [24]. Based on SIFT, A. Sedaghat proposes Uniform Robust Scale Invariant Feature Transform (UR-SIFT), which improves the distribution quality of SIFT in image space and scale [25]. G.-R. Cai proposed the Perspective Scale Invariant Feature Transform (P-SIFT), which simulates the deformation of the scene in the multi-view image through the perspective sampling of the virtual camera, which effectively improves the robustness of the algorithm to the perspective change [26]. In order to address the problem of geometric distortion in remote sensing images, A. Sedaghat proposed Adaptive Binning Scale Invariant Feature Transform (AB-SIFT). According to the adaptive histogram quantization strategy, Hesse affine algorithm descriptor is utilized to make the local descriptor highly unique and robust to geometric distortion and radiation distortion [27]. Scale Invariant Feature Transform (SIFT) class descriptors have high matching precision and are robust to scale, rotation and illumination, but even SURF is still difficult to satisfy the real-time needs. The binary descriptor came into being to reduce the computational cost and improve the matching efficiency. Because the binary descriptor uses hamming distance for matching and image pixel pairs for comparison, it has considerable advantages in memory use and matching speed. Binary Robust Independent Elementary Features (BRIEF) [28] uses a fixed 9 * 9 smooth convolution kernel, which has faster descriptor establishment speed, but does not have rotation invariance. Fast Retina Key point (FREAK) [29] adopts a sampling mode which is close to the image information received by the human retina, which has the advantages of fast calculation and small memory consumption. Binary Online Learning Descriptor (BOLD) [30] is independently optimized for each image block to get a more robust descriptor. At present, BinBoost [31] and LATCH [32] are the most stable. Based on SIFT and orientation function analysis system technology, Y. Zhang improves ORB algorithm [33]. This algorithm realizes large-size ultra-high resolution image registration algorithm from coarse to fine, and accelerates the acquisition of feature points and image correction [34]. Center-Symmetric Local Binary Pattern (CS-LBP) [35] proposed by M. Heikkila is a variant of SIFT and local binary. Boosted Efficient Local Image Descriptor (BELID) [36] proposed by I. Suárez utilizes the integral image calculates the difference of the average gray value of the square region of the image. This method improves the calculation speed, the precision is comparable to that of SIFT, and the running speed is close to ORB. In order to further improve the computational efficiency, I. Uárez proposes Boosted Efficient Binary Local Image Descriptor (BEBLID) [37] based on BELID. It uses AdaBoost to improve the feature selection process of BELID, gets better local matching, and is more efficient than ORB. To generate key points efficiently, S. Leutenegger proposes Binary Robust Invariant Scalable Key points (BRISK) [38], which greatly reduces the computational cost.

Based on BRISK, C. H. Tsai improves the running speed again, and proposes Accelerated Binary Robust Invariant Scalable Key points (ABRISK) [39]. Uncorrected feature pairs are screened out as early as possible by sorting ring, which effectively improves the matching logarithm after image enhancement processing. The author verifies that this method is insensitive robust to image size changes and radiation changes. Based on ABRISK and combining the distribution of human retinal ganglion cells and visual regulation, M. L. Cheng proposes a Inverse Sorting Ring (ISR), which rearranges ABRISK, and proposes Enhanced Accelerated Binary Robust Invariant Scalable Key points (EABRISK) [40], which improves the matching precision of high similarity pixel intensity. To improve the robustness of image features, the number of matching and the efficiency of data processing again, M. L. Cheng adds color information to the descriptor. Based on EABRISK, Synthetic-Colored Enhanced Accelerated Binary Robust Invariant Scalable Key point (SC-EABRISK) [41] is proposed, which runs 20 times faster than EABRISK.

The region-based method has high matching precision, but the computational cost is high. Among the feature-based methods, SIFT matching method is robust to illumination, scale, rotation and so on, but its running speed is slow. Binary matching method sacrifices the matching precision in exchange for higher algorithm efficiency. Even with the continuous development of region-based and feature-based methods, however, remote sensing image registration still has the following difficulties: 1. diversity of sensor data types and conditions during data acquisition; 2. data size; 3. lack of known image model; 4. lack of uniformly distributed feature datum points. The ways to get remote sensing images mainly include satellite camera, drone camera, aerial camera and so on. Aerial camera has become the main means to get geographic information because of its strong flexibility and high efficiency of getting information. In the process of continuously getting aerial remote sensing images, the swing-sweep type aerial camera expands the field of view, but brings significant local geometric distortion to the aerial remote sensing images because of the change of camera swinging angle. For the images in the same swinging strip, the larger the swinging angle is, the higher the degree of local geometric distortion is. Although the method of Gaussian weighting function, which is typically used, can reduce the influence of local geometric distortion to some extent, it reduces the significance of the descriptor. At present, by using modern remote sensing sensors and navigation devices such as Global Position System (GPS) and Inertial Navigation System (INS), registration can be achieved by direct geographic reference. This registration method can eliminate obvious global geometric distortion, but dozens of pixels will still be offset by this method, and the overall registration effect is poor [42–44]. However, multitime effect and terrain fluctuation will lead to the registration error because of the radiation and geometric differences between the images to be registered. At present, region-based methods and feature-based methods often have many mismatching in image blur, which can't be applied to feature matching between aerial images with different sharpness.

Therefore, this paper proposes a feature matching algorithm between aerial images with different sharpness by using DEM data and multiple constraints. In this paper, to solve the problem of pixel offset in registration using GPS and INS, lie group is used to represent the relative motion and coordinate transformation of aerial camera, and the aerial imaging model is modified to reduce pixel offset based on eliminating global geometric distortion. However, the modified aerial imaging model still contains residual errors, which can't accurately confirm the location of feature matching points, and if the nearest neighbor algorithm is utilized directly according to the pixel coordinate position, it will cause about 27.43% mismatching. Therefore, in order to better determine the search location of feature matching points, nonlinear soft margin support vector machine is utilized to expand the modified aerial imaging model, so point-to-point matching is extended to point-to-surface matching to make the

surface become the search range of the feature matching point. To further determine the location of feature points, the epipolar geometric constraint is introduced to reduce the search range from surface search to line search. The L-k optical flow method after velocity compensation is used to determine the most suitable location of feature matching points. To reduce the registration error caused by multitime effect and terrain relief, this algorithm introduces DEM data with millimeter accuracy. In this paper, the feature extraction algorithm uses the ABRISK to meet the real-time needs. The feature extraction algorithm runs fast and is insensitive to the change of image size, and can process large aerial remote sensing images got by aerial cameras in real time.

The contribution of this work is outlined below. The "Algorithm Flow" section will briefly introduce the overall flow of the algorithm of this paper. In the "Optimization of Imaging Model" section, the special Euclidean group in Riemannian geometry and the concept of non-linear soft margin support vector machine in machine learning are used to optimize the aerial imaging model and extend the feature matching points to the search range of surfaces. The "Feature matching algorithm" section introduces two kinds of constraints used in the feature point extraction algorithm: polar constraint and extended L-k optical flow, and combines it with the aerial imaging model to reduce the search range of the surface to the line search range to determine the most appropriate feature matching position. In "Results" section, experiments are carried out to verify the algorithm in this paper. The first group of experiments verifies the rapidity of the feature matching algorithm when the sharpness of aerial image is similar. Compared with the classical SURF algorithm, ORB algorithm, BRISK algorithm, ABRISK algorithm and geographic information algorithm, it is proved the running speed of the text algorithm can fully meet the real-time needs after being accelerated by FPGA hardware. The second group of experiments compares the extraction effects of classical feature extraction algorithms such as FAST, SURF, ORB and BRISK on blur aerial images, and discusses the reasons for using ABRISK algorithm to extract feature points in this paper. The third group of experiments compares the feature matching effects of this algorithm with the classical BRIEF, SURF and BRISK algorithms in different sharpness images. The fourth group of experiments to verify the repeatability of this algorithm, repeated experiments on this algorithm and a variety of classical algorithms, and calculated its matching precision. The fifth group of experiments tested the impact of DEM image accuracy on this algorithm. The experimental results show that the algorithm matches the features of aerial images with different sharpness, and the matching precision can reach 90%. This shows the algorithm in this paper can achieve high matching precision between two aerial remote sensing images with different sharpness, and the running speed of the algorithm meets the real-time needs. The "Conclusions" section gives the relevant conclusions.

## Materials and methods

### Algorithm flow

Based on the aerial camera calibrating the inner orientation elements in the laboratory and getting the high-accuracy DEM image of the aerial camera photographing area in advance, the algorithm in this paper takes the back-end model of aerial photogrammetry as a known parameter and loads it into the preprocessing process. The specific process of the algorithm of this paper is as follows: the swing-sweep type aerial camera takes the previous image, and according to the position and attitude information of the previous image simultaneously got by GPS and INS, ABRISK algorithm is used to extract the feature points and calculate the descriptor accordingly. The extracted feature points are obtained through the back-end modified aerial imaging model and the elevation information provided by high-accuracy DEM to

get the geographical coordinates of the spatial object points. And try to use ABRISK algorithm to extract feature points while photographing the latter image with swing-sweep type aerial camera. If the feature matching is successful, the latter image is clear enough and does not belong to the scope of this algorithm. Only according to the attitude and position information of the aerial camera provided by the aircraft orientation and positioning system, calculate the coordinate position of the latter aerial image, search the feature points of ABRISK algorithm, and complete the feature matching algorithm. If ABRISK algorithm fails to complete feature matching, then the latter aerial image is not clear enough to use conventional algorithms to achieve feature matching, using this algorithm for feature matching. That is, the point matching is extended to the area search range by using the nonlinear soft margin support vector machine extended modified aerial imaging model, and according to the epipolar geometric constraints formed by the swing-sweep type aerial camera, the matching range of feature points is reduced from the area search range to the line search range. Then the extended L-k optical flow method is used to speculate the position of the previous feature points in the latter aerial image, and the modified feature matching search point is used to realize the feature matching algorithm between different sharpness images. The precise algorithm flow is shown in Fig 1.

## Optimization of imaging model

Using aerial imaging model for remote sensing image registration can eliminate the global geometric distortion, and the registration error is usually about dozens of pixels, which is difficult to meet the precision requirements of aerial camera sharpness detection. Therefore, this paper uses the special Euclidean group in Lie group to represent the relative motion and coordinate transformation of aerial camera, and modifies it according to the position and attitude information got by GPS and INS. The modified aerial imaging model improves the feature matching effect of aerial remote sensing images. However, in the process of getting aerial remote sensing images, in addition to the change of aircraft attitude and ground elevation, there is also a change of swinging photography angle perpendicular to the flight direction of the aircraft, resulting in having local geometric distortion of the got image. The distortion reduces the feature matching precision, so the nonlinear soft margin support vector machine is introduced to expand the search range of the feature points got by the aerial imaging model, so the most appropriate search range is selected according to the different positions of the camera in the swinging strip, to improve the matching precision and efficiency, and reduce the influence of aerial camera swinging angle on feature matching precision.

**Lie group representation of camera motion.** There will be relative motion between the two images taken by the swing-sweep type aerial camera, which includes not only the change of the relative position of the camera caused by the change of the attitude and position of the aircraft, but also the change of the swinging angle of the swing-sweep type aerial camera. The motion of the swing-sweep type aerial camera is complex. In this paper, Lie group is utilized to represent the relative motion of the camera before and after photographing, which can optimize the imaging model and combine with nonlinear soft margin support vector machine. Thus the feature offset caused by the residual error which is not completely removed by the modified aerial imaging model is tolerated to a certain extent. The local geometric distortions of aerial remote sensing images got by swing-sweep type aerial cameras at different swinging positions in the same swinging strip are different. Therefore, the relaxation factor in nonlinear soft margin support vector machine is utilized to expand the search range of aerial imaging model feature points, and the point correspondence is extended to surface correspondence.

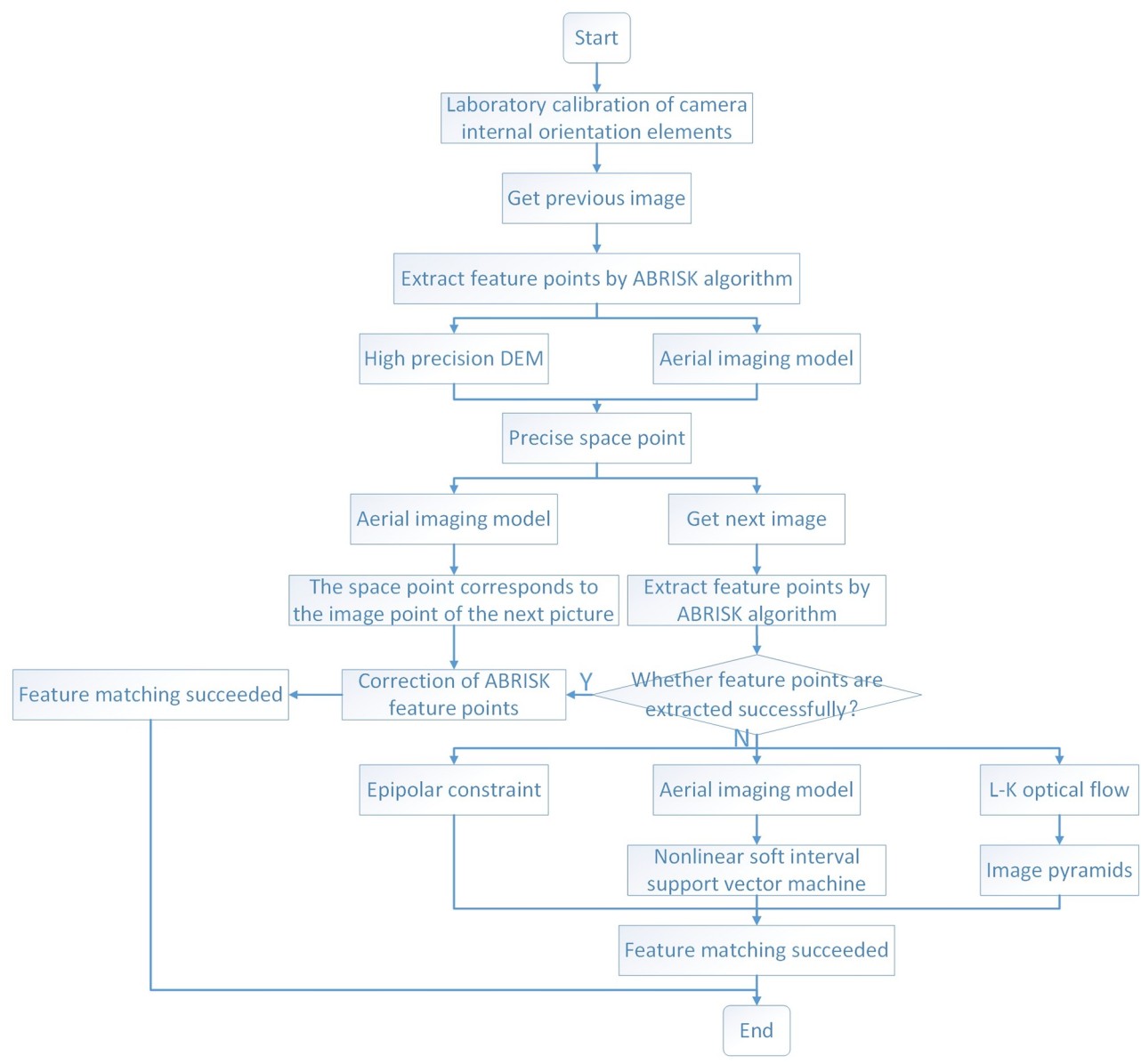

**Fig 1. Overall flow chart of algorithm.**

The establishment of aerial camera imaging model is usually based on the optical imaging principle, that is, the ground object point, photography center and the image point are located in the same straight line. Homogeneous coordinates are used to represent the positions of ground object points, photography centers and image points in space, and the relative coordinate transformation relations of the earth-centered earth-fixed coordinate system, the geodetic coordinate system, the aircraft coordinate system and the photographic coordinate system are established. It is represented by the special Euclidean group in Riemannian geometry, so the aerial imaging model of swing-sweep type aerial camera can be modified by differential manifold.

The camera coordinate position is described by the WGS84 earth ellipsoid model the earth-centered earth-fixed coordinate system $O - X_i Y_i Z_i$ [45], as shown in Fig 2. The coordinate of

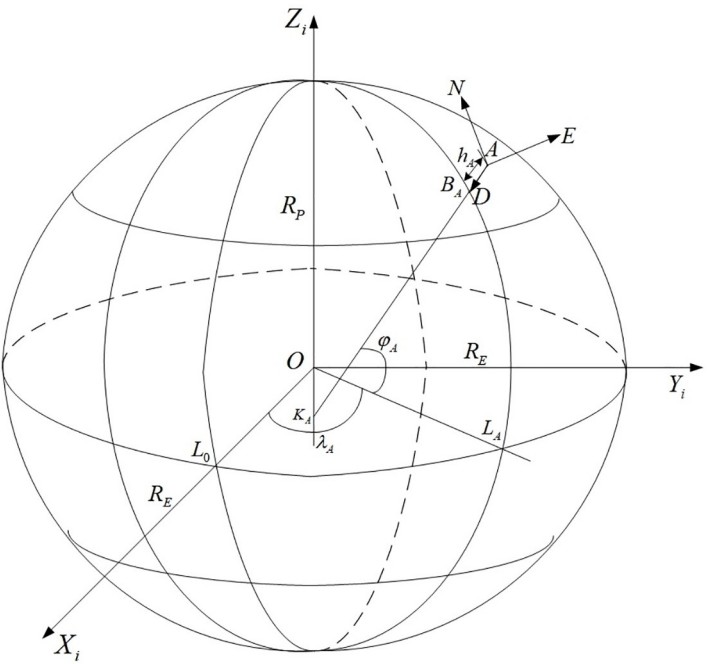

**Fig 2. The WGS84 earth ellipsoid model.**

the photography center $S_1$ in the earth-centered earth-fixed coordinate system $O − X_i Y_i Z_i$ of the swing-sweep type aerial camera photographing the previous image is $(X_{S1}^i, Y_{S1}^i, Z_{S1}^i)$, and the coordinates of the photography center $S_2$ in the earth-centered earth-fixed coordinate system $O − X_i Y_i Z_i$ photographing the latter image is $(X_{S2}^i, Y_{S2}^i, Z_{S2}^i)$, as shown in Fig 3. The relative translation motion of the aerial camera is because of the change of the position of the aircraft in the two images previous and latter photographing, then the change of the position of the

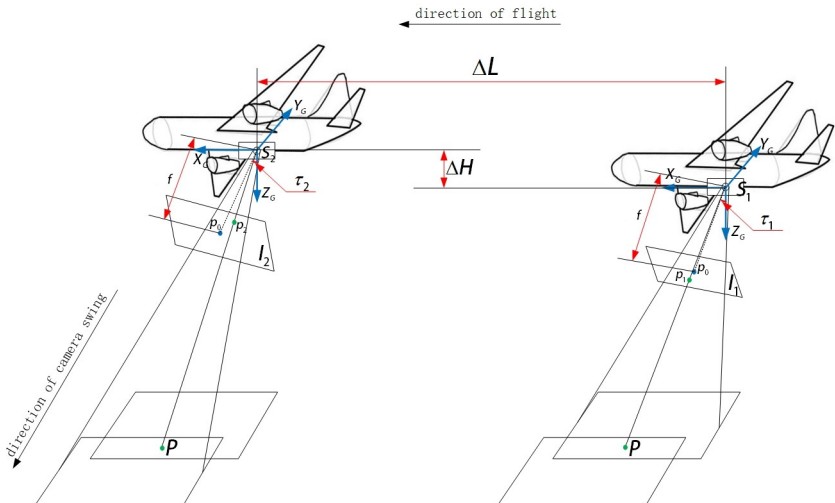

**Fig 3. Schematic diagram of position changes of two images between the previous and latter aerial images photographing by swing-sweep type aerial camera.**

aircraft can be regarded as the translation of the camera, using t to represent [46], there is:

$$t = \begin{bmatrix} \Delta X_{21}^i \\ \Delta Y_{21}^i \\ \Delta Z_{21}^i \end{bmatrix} = \begin{bmatrix} \Delta X_{S2}^i \\ \Delta Y_{S2}^i \\ \Delta Z_{S2}^i \end{bmatrix} - \begin{bmatrix} \Delta X_{S1}^i \\ \Delta Y_{S1}^i \\ \Delta Z_{S1}^i \end{bmatrix}$$

$$t^\wedge = \begin{bmatrix} 0 & -\Delta X_{21}^i & \Delta Y_{21}^i \\ \Delta X_{21}^i & 0 & -\Delta Z_{21}^i \\ -\Delta Y_{21}^i & \Delta Z_{21}^i & 0 \end{bmatrix}, \tag{1}$$

The relative rotation motion of the swing-sweep type aerial camera between the previous and latter aerial images photographing can be regarded as two parts, one part is the attitude of the aircraft has changed between the previous and latter aerial images photographing, and the other part is the swing-sweep type aerial camera itself will increase a certain swinging angle between the previous and latter aerial images to expand the range of images. That is, the relative rotational motion of the camera can be expressed as:

$$R = \begin{bmatrix} 1 & 0 & 0 \\ 0 & \cos(\Delta\psi + \Delta\tau) & \sin(\Delta\psi + \Delta\tau) \\ 0 & -\sin(\Delta\psi + \Delta\tau) & \cos(\Delta\psi + \Delta\tau) \end{bmatrix}$$

$$\begin{bmatrix} \cos(\Delta\omega) & 0 & -\sin(\Delta\omega) \\ 0 & 1 & 0 \\ \sin(\Delta\omega) & 0 & \cos(\Delta\omega) \end{bmatrix} \begin{bmatrix} \cos(\Delta\kappa) & \sin(\Delta\kappa) & 0 \\ -\sin(\Delta\kappa) & \cos(\Delta\kappa) & 0 \\ 0 & 0 & 1 \end{bmatrix}, \tag{2}$$

Where, $\Delta\psi = \psi_2 - \psi_1$ is the change of aircraft attitude roll angle between the previous and latter aerial images photographing, $\Delta\omega = \omega_2 - \omega_1$ is the change of aircraft attitude pitch angle between the previous and latter aerial images photographing, $\Delta\kappa = \kappa_2 - \kappa_1$ is the change of aircraft heading angle between the previous and latter aerial images photographing, $\Delta\tau = \tau_2 - \tau_1$ is the change of swinging angle between the previous and latter aerial images photographing by swing-sweep type aerial camera photographing. Compared with the UAV camera and the push-broom aerial camera, the relative rotation motion of the camera in addition to the change of the attitude of the aircraft in the two aerial remote sensing images got by the swing-sweep type aerial camera in the same swinging strip, it also includes the change of camera swinging angle in the aerial remote sensing image. If the change of the angle is not measured and corrected, it will cause serious local geometric distortion of the aerial remote sensing image and reduce the matching precision. The camera relative motion relationship between the two aerial remote sensing images got by the aerial camera in the same swinging strip is shown in Fig 3. In Fig 3, when the aerial camera gets the previous aerial remote sensing image, the photography center is located at $S_1$. The corresponding image point of the ground scene point P in the image plane $I_1$ is $p_1$, the aircraft attitude is $(\psi_1, \omega_1, \kappa_1)$, and the camera swinging

angle is $\tau_1$. After getting the image interval time $\Delta T$, at this time, the aircraft moves $\Delta L$ in space from the position where the previous aerial remote sensing image is obtained, and the height moves $\Delta H$, and the photography center is located at $S_2$, and the corresponding image point of the ground scene point P in the image plane $I_2$ is $p_2$, the attitude of the aircraft is $(\psi_2, \omega_2, \kappa_2)$, and the camera swinging angle is $\tau_2$. It is easy to know the relative translation motion t and relative rotation motion R can be expressed by Eqs (1) and (2) between the previous and latter aerial images acquisition.

Given the relative translation motion t and relative rotation motion R in the two images between the previous and latter aerial images by aerial camera photographing, the relative motion of the camera can be written in the form of special Euclidean group SE(3) [47, 48], that is:

$$SE(3) = \begin{bmatrix} R & t \\ 0 & 1 \end{bmatrix}, \tag{3}$$

**Nonlinear soft margin support vector machine.**    The basic form of aerial camera imaging model based on the pinhole model is as follows:

$$\frac{x_{p_2}^i - X_{S_2}^i}{X_P^i - X_{S_2}^i} = \frac{y_{p_2}^i - Y_{S_2}^i}{Y_P^i - Y_{S_2}^i} = \frac{z_{p_2}^i - Z_{S_2}^i}{Z_P^i - Z_{S_2}^i}, \tag{4}$$

Where, $(x_{p_2}^i, y_{p_2}^i, z_{p_2}^i)$ is the coordinates of the corresponding image point $p_2$ got by the aerial imaging model on the earth-centered earth-fixed coordinate system $O - X_iY_iZ_i$, and $(X_{S_2}^i, Y_{S_2}^i, Z_{S_2}^i)$ is the coordinate of the aerial camera photography center $S_2$ got by GPS and INS on the earth-centered earth-fixed coordinate system $O - X_iY_iZ_i$ coordinates. $(X_P^i, Y_P^i, Z_P^i)$ is the coordinate of the precise object point got from the high-accuracy DEM image on the earth-centered earth-fixed coordinate system $O - X_iY_iZ_i$.

When there is an error in the attitude or position data such as the swinging angle of the aerial camera, the Eq (4) will be changed to:

$$\frac{x_{p_2'}^i - X_{S_2'}^i}{X_P^i - X_{S_2'}^i} = \frac{y_{p_2'}^i - Y_{S_2'}^i}{Y_P^i - Y_{S_2'}^i} = \frac{z_{p_2'}^i - Z_{S_2'}^i}{Z_P^i - Z_{S_2'}^i}, \tag{5}$$

Where, $(x_{p_2'}^i, y_{p_2'}^i, z_{p_2'}^i)$ is the coordinates of the high quality feature point $p_2'$ got by the aerial camera using the ABRISK algorithm on the earth-centered earth-fixed coordinate system $O - X_iY_iZ_i$, and $(X_{S_2'}^i, Y_{S_2'}^i, Z_{S_2'}^i)$ is the real position coordinates of the aerial camera photography center $S_2$ in the earth-centered earth-fixed coordinate system $O - X_iY_iZ_i$.

Using aerial imaging model for feature matching can eliminate global geometric distortion, but the matching effect is poor, usually with a relative offset of about dozens of pixels. In the process of getting the aerial remote sensing image, the relative rotation motion of the swing-sweep type aerial camera not only changes the attitude of the aircraft, but also changes the relative swinging angle, which results in the local geometric distortion of the got aerial remote sensing image. The relative offset of the pixel will be increased, and the matching effect will be even worse. Although the position of the feature points got by the modified aerial camera imaging model removes the influence of external reasons such as temperature and air pressure, there will still be a relative pixel offset because of the influence of residual error. In the experiment, it is found that 27.43% of the features will be mismatched if the nearest neighbor algorithm is directly used. Therefore, according to the different geometric distortion characteristics

of the aerial remote sensing images with different swinging positions in the swinging strip, the nonlinear soft margin support vector machine is used to expand the position of the feature points located by the modified aerial imaging model. The point correspondence is extended to the face correspondence. And abandon the inherent nearest neighbor location method, and use the relaxation factors of different positions corresponding to the internal feature position as the basis for calculating the search range of feature points. Support vector machine is a traditional machine learning classification algorithm [49], which is mainly suitable for linear binary classification problems. Compared with the support vector machine, the soft interval support vector machine has a certain fault tolerance due to the use of relaxation factor. In the two-classification problem, there are few models that accord with linear separability. In the process of application, kernel techniques are usually used to improve computational efficiency. The basic representation of the nonlinear soft margin support vector machine is:

$$\min_{\gamma,b} \frac{1}{2}\|\gamma\|^2 + C\sum_{i=1}^{N}\xi_i \quad s.t. \quad y_i[\gamma^T\phi(X_i+b)] \geq 1-\xi_i \quad \xi_i \geq 0, \tag{6}$$

Where, $y_i(i = 1, 2, \cdots, n)$ is the label of the feature point, and it is 1 when it is determined to be the feature point, otherwise it is 0. $\gamma$ is the vector parameter, $X_i(i = 1, 2, \cdots, n) = \{x_{i1}, \cdots, x_{im}\}$ is the feature point, $x_{i1}, \cdots, x_{im}$ is the characteristic factor of the feature point, b is the vector parameter, $\xi_i(i = 1, 2, \cdots, 12)$ is the relaxation factor, C is a constant, and $C > 0$, $\phi(X_i)$ chooses the radial basis function kernel function, that is:

$$\theta\left(X_j, X_k\right) = e^{-\frac{\|X_j - X_k\|^2}{2\sigma^2}}, \tag{7}$$

The schematic diagram of the extended and modified aerial imaging model using nonlinear soft margin support vector machine is shown in Fig 4. In Fig 4, the position of the feature point located by the aerial imaging model is $p_2$, and the position of the feature point using the nearest neighbor matching is the mismatching point. Therefore, according to the position of the actual feature point $p_2'$, the appropriate relaxation vector is obtained, which is used in the nonlinear soft margin support vector machine, and the aerial imaging model is extended to determine the search position of the feature point.

The image registration algorithm based on geographic information can't make up for this error because it ignores the rich image information, so it can't be optimized to further improve its precision.

When the sharpness of the two images is the same, the feature points are extracted by ABRISK algorithm, and then matched by the aerial imaging model. Then the characteristic factor vector of the feature point is set to $[x_{i1}, \cdots, x_{im}] = [x_{p_2'}^i, y_{p_2'}^i, z_{p_2'}^i]$, and the parameters can be solved by nonlinear soft margin support vector machine and Stochastic Gradient Descent method (SGD) [50]. Thus, the point correspondence of the modified aerial imaging model is extended to face correspondence by nonlinear soft margin support vector machine, and then the surface search is reduced to line search by adding epipolar geometric constraints. Finally, the most suitable feature point position is determined by extending the L-k optical flow, as described below.

## Feature matching algorithm

**Epipolar constraint.** Epipolar constraint is a common algorithm for camera motion estimation after 2D-2D image registration in visual Simultaneous Localization And Mapping

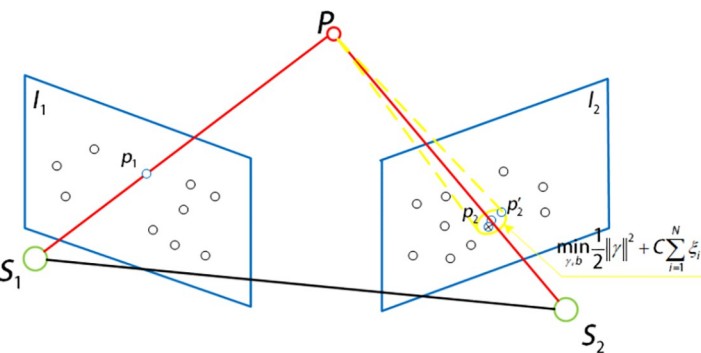

**Fig 4. Aeronautical imaging model with soft margin support vector machine.**

(SLAM) [51]. It has scale uncertainty because of the uncertainty of spatial point P position, so it is one of the soft constraints.

For the swing-sweep type aerial camera, the camera position includes not only translation, but also the change of camera swinging angle and aircraft attitude between the previous and latter aerial images photographing, so it has the epipolar geometric constraints the vertical push-broom aerial camera does not have. A small range of surface search is determined according to the relaxation factor in the nonlinear soft margin support vector machine and the point position determined by the modified aerial imaging model, which can give the possibility of mismatching point pair correction to a certain extent. However, because the point correspondence is extended to the area search scope, if the search is traversed by violence, the computing time will be greatly increased. Therefore, the dimension reduction search of the epipolar geometric constraint of the swing-sweep type aerial camera in the aerial remote sensing image is used to reduce the area search range to the line search range, which greatly reduces the operation time and improves the efficiency of feature location determination. The epipolar constraint generation diagram of the swing-sweep type aerial camera is shown in Fig 5. In Fig 5, when the swing-sweep type aerial camera gets the previous aerial remote sensing image, the photography center is located at $S_1$. At this time, the swinging angle of the aerial

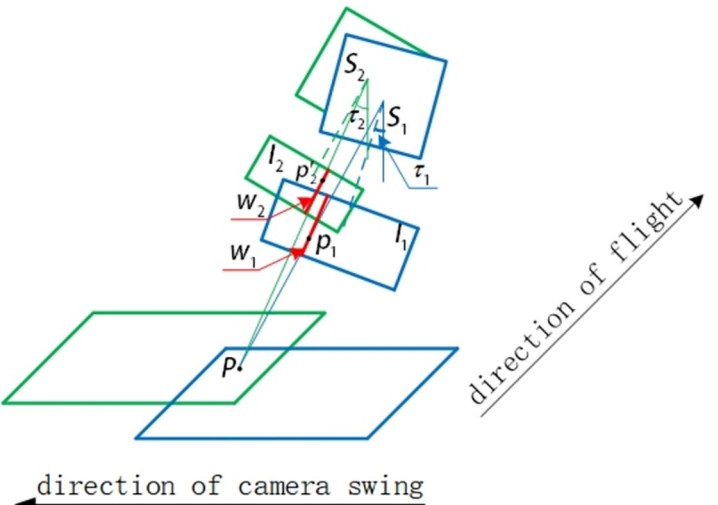

**Fig 5. Schematic diagram of epipolar constraint of swing-sweep type aerial camera.**

camera is $\tau_1$, and the corresponding image point of the ground object point P on the image plane $I_1$ using the data provided by the high-accuracy DEM image is $p_1$. In the two images got by the swing-sweep type aerial camera, there is a change in the relative translation motion of the camera and the swinging angle, so the epipolar will be produced on the image plane. After the interval time $\Delta T$ of image acquisition, the photography center is moved to $S_2$, and the aerial camera is swinging to $\tau_2$. According to the spatial position of the swing-sweep type aerial camera and the ground object point in getting the previous aerial image, the epipolar $W_2$ is generated on the image plane $I_2$. According to the epipolar geometric relationship, the corresponding image point $p_2'$ of the ground object point P on the image plane $I_2$ must be located on the epipolar $W_2$.

The position of aerial camera photography center $S_2$ is reached by camera relative motion through photography center $S_1$. The relative motion of the camera is expressed by a special Euclidean group SE(3), such as Eq (3). Then the classical Rodriguez formula can be used to realize the transformation between the corresponding special Euclidean group SE(3) and its lie algebra se(3).

The transformation of a special Euclidean group SE(3) into its lie algebra se(3) can be derived by a perturbed model, then the epipolar geometric constraint can be expressed as:

$$p_2^T K^{-T}(R + dR)K^{-1}p_1 = 0, \tag{8}$$

Where the feature point $p_2'$ is the pixel coordinate on the image plane $I_2$, the calibration matrix is $K = \begin{bmatrix} f_x & s & c_x \\ 0 & f_y & c_y \\ 0 & 0 & 1 \end{bmatrix}$, and the camera translation matrix $t\wedge$ is shown in Eq (1), and the feature point $p_1$ is the pixel coordinate on the image plane $I_1$.

When the sharpness of two aerial images is the same, the minimum dR of Eq (8) is calculated iteratively, which is regarded as a fixed value, and when matching feature points in images with insufficient clarity, it is directly introduced as a correction.

The epipolar geometric constraint can reduce the search range of feature points determined by the extended aerial imaging model of nonlinear soft margin support vector machine from surface search to line search, to reduce the search range of feature points and improve the efficiency of feature point matching. And it is not affected by the image sharpness. Therefore, in the feature matching between different images, it can increase the precision and efficiency of feature matching as geometric constraints. Fig 6 is schematic diagrams of the form of epipolar geometric constraints on the image plane and their combination with the aerial imaging model extended by nonlinear soft margin support vector machine, respectively.

**Extended L-K optical flow.**   L-k optical flow method [52] is a common method in visual odometer in visual SLAM based on the strong assumption: gray constant assumption. However, the use of L-k optical flow method should include three premises: (1) the assumption of constant luminance: the assumption the brightness of the feature points of the same object should be constant in the two images taken by the camera; (2) the assumption of time continuity: the objects in the image move slowly or remain the same; (3) the spatial consistency hypothesis: the motion of the adjacent pixels in the image is consistent.

Aerial camera as a common means to get ground information, aerial images also have their unique advantages, the photographing interval is short, and most of the objects are still scenes on the ground, which meet the conditions of L-k optical flow method.

The ground scenes captured by aerial images are similar, and their gray scale is also close. If gray scale is still used as the condition of L-k optical flow extraction, it will cause many matching errors. Therefore, this paper calculates the optical flow according to the high-accuracy

(A)

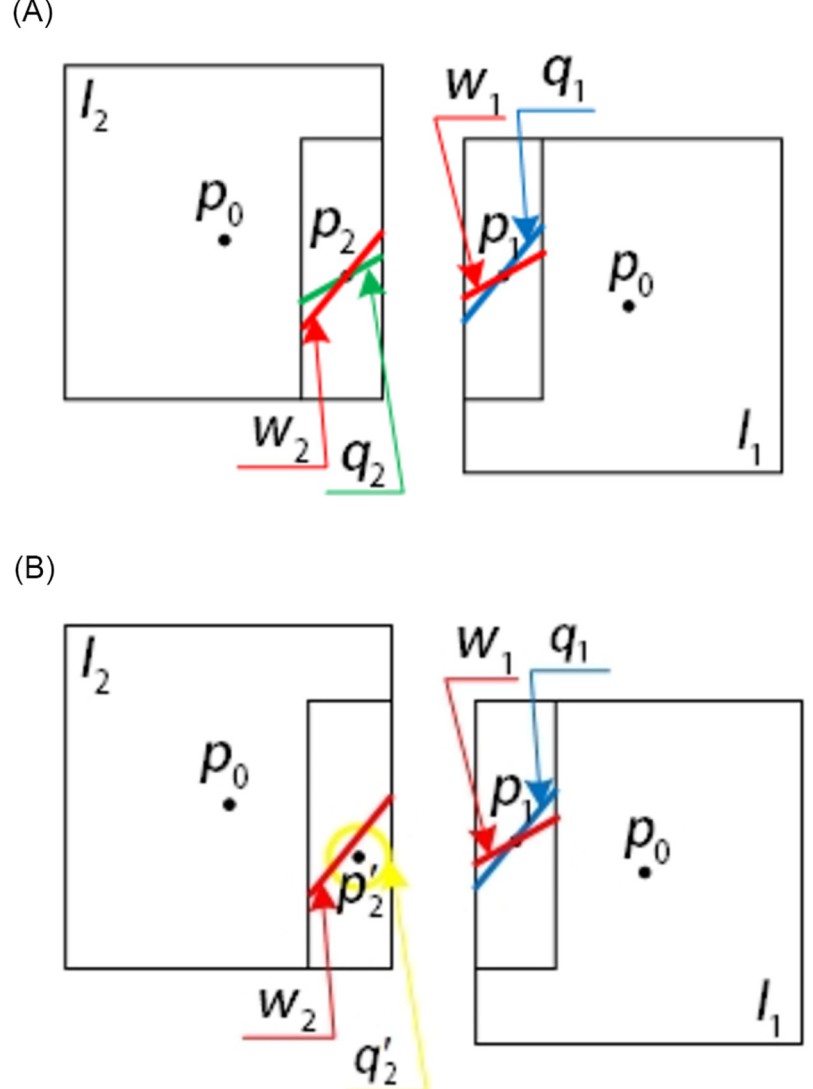

(B)

**Fig 6. Form of image epipolar constraint on image plane.**

DEM modified feature points, that is:

$$J(m + dm, n + dn, t + dt) = J(m, n, t), \tag{9}$$

Where, $(m, n)$ is the pixel coordinates of the feature points extracted by the ABRISK algorithm for the pervious aerial image of the aerial camera, T is the time of the pervious aerial image of the aerial camera, and dT is the time interval between the pervious aerial image and the latter aerial image of the aerial camera. The J function can use ABRISK feature points to get the coordinates of ground points. As shown in Fig 7.

In little difference in the sharpness of two aerial images, the matching of corresponding feature points in different images is obtained by ABRISK algorithm, and after the matching is modified based on geographic information, the pixel moving speed can be obtained by using

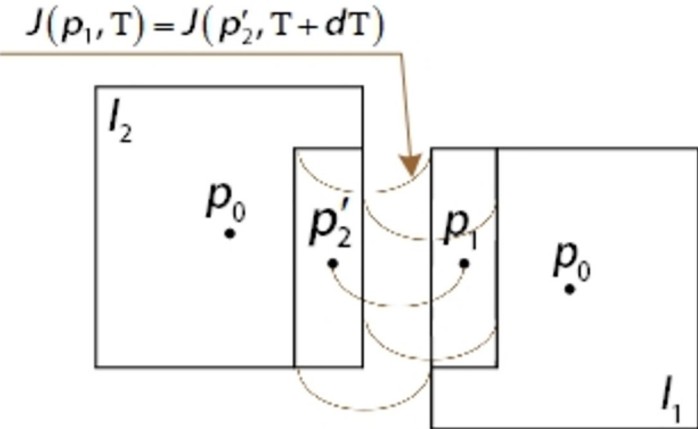

**Fig 7. Schematic diagram of optical flow method between aerial camera swing scanning images.**

least square method:

$$\begin{bmatrix} u \\ v \end{bmatrix} = -(A^T A)^{-1} A^T b \quad A = \begin{bmatrix} \dfrac{\partial J}{\partial x_1} & \dfrac{\partial J}{\partial y_1} \\ \vdots & \vdots \\ \dfrac{\partial J}{\partial x_r} & \dfrac{\partial J}{\partial y_r} \end{bmatrix} \quad b = \begin{bmatrix} \dfrac{\partial J}{\partial t_1} \\ \vdots \\ \dfrac{\partial J}{\partial t_r} \end{bmatrix}, \tag{10}$$

Because the algorithm in this paper is based on the swing-sweep type aerial camera design, the swinging angle of the swing-sweep type aerial camera changes, so the pixel moving speed will change. For example, if the relative motion of the pixel is regarded as a function of time T, the calculation will be greatly complicated. Therefore, this paper adopts different compensation for the aerial images taken by different groups of aerial cameras, that is, between the swinging strips. The pixel coordinates position between the ith image and the i+1th image can be calculated using the following formula:

$$J(m'_{i+1}, n'_{i+1}, t + dt) = J(m_i + (u + u_i)dt, n_i + (v + v_i)dt, t), \tag{11}$$

## Summary of feature matching algorithms between aerial images with different sharpness

In the process of aerial camera photographing, the changes of external environment, such as pressure, temperature and ground elevation difference, lead to the difference of aerial image sharpness between the pervious and the latter of the camera photographing. In blur image, the method of using feature point offset to calculate the sharpness detection of aerial image will fail because of high mismatch rate. To resist the influence of pressure, temperature and other reasons, this algorithm uses a special Euclidean group to represent the relative motion of the camera and modifies the imaging model. To resist the influence of ground elevation difference, the algorithm uses high-accuracy DEM image to extract the corresponding ground point information. In blur image, both region-based method and feature-based method will cause poor matching effect and even unable to complete feature matching. Therefore, this algorithm

uses ABRISK to extract feature points on the clear image, and utilizes the modified aerial imaging model to determine the location of the feature points on the blur image. Because only the modified aerial imaging model is used for feature matching, the local geometric distortion can't be removed, and the pixel offset of feature points will be caused by the residual error. Therefore, it is unreliable to use only the modified aerial imaging model for feature matching, especially in the aerial images with large swinging angle in the swinging strip. In this paper, the extended modified aerial imaging model of nonlinear soft margin support vector machine is adopted. According to aerial images got from different swinging angles, different relaxation factors are obtained, and the matching search range of feature points is expanded. To the maximum extent, in ensuring the matching precision, minimize the area search range. If traversing every point in the surface search range, the running time of the algorithm is too slow, and the location of the feature points can't be determined. In this paper, the epipolar geometric constraint is used to reduce the surface search range to the line search range, which improves the search efficiency and precision. However, it still can't meet the real-time needs of the algorithm. In this paper, extended L-k optical flow is used to compensate the relative movement speed of pixels in different swinging positions in the swinging strip. According to the interval time of the image photographing, the pixel movement is obtained, and the best feature matching position is determined by integrating the epipolar geometric constraints.

The overall schematic diagram of the image feature matching algorithm with different sharpness is shown in Fig 8. Using the ABRISK algorithm to get the feature point $p_1$ in the previous image, the coordinates of the spatial point P are calculated by the aerial imaging model and high-accuracy DEM, and the attitude information got by GPS and INS is used to get the relative motion of the camera and converted into the form of special Euclidean group orientation SE(3). Based on the coordinate information of spatial points, the feature matching points that best meet the needs are obtained by nonlinear soft margin support vector machine, polar constraint and extended L-k optical flow method. That is:

$$\min_{p_2} p_2^T K^{-T} t^\wedge (R + dR_i) K^{-1} p_1 + \| p_2 - [p_1 + (v + v_i)dt] \|^2$$

$$s.t. \quad \gamma^T \phi(p_2) + b - 1 + \xi_i \geq 0,$$

(12)

Where, $p_2$ is the pixel coordinate of the feature point determined by the aerial imaging model Eq (5); K is the calibration matrix; $t^\wedge$ is the antisymmetric matrix of the relative translation motion vector between the two aerial images of the aerial camera pervious and latter photographing, see Eq (1); R is the relative rotational motion matrix between the two aerial images pervious and latter aerial camera photographing. $dR_i$ is the relative rotation compensation matrix between the ith image and the i+1th image determined by Eq (8) when the sharpness is similar; $p_1$ is the exact pixel coordinates of the feature points got by the ABRISK algorithm in the front aerial image of the aerial camera; $v$ is the basic speed of the pixel movement of the feature points of the aerial camera between the two images; $v_i$ is the compensation speed of the pixel motion of the feature points of the aerial camera between the ith image and the i+1th image. dT is the amount of time change between the two aerial images taken by the aerial camera; $\gamma$, $b$ is the parameter vector got by using the feature point matching pair Eq (6) when the sharpness of the image taken by the aerial camera is similar; and $\xi_i$ is the relaxation factor.

The feature matching points are obtained by solving Eq (12) to complete the task of feature matching between aerial remote sensing images with inconsistent sharpness.

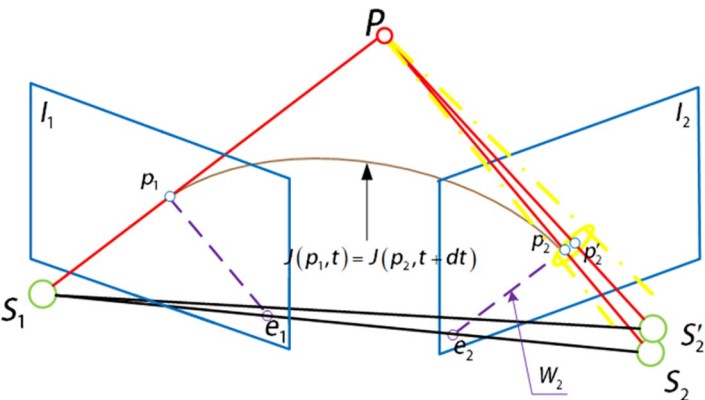

**Fig 8. General diagram of feature matching algorithm.**

## Results

In this paper, the tilt aerial image taken by the national high score special XX tilt measuring camera on August 1st, 2021 is tested. The algorithm running computer uses the CPU for Intel Core i7–8900X, the programming language is C++, and the OpenCV data packet is used.

The area captured by aerial remote sensing images captured by aerial cameras has a high similarity, especially when getting images of farmland, urban buildings and other areas. Using feature-based method for feature matching algorithm will seriously reduce the success rate of feature matching.

To discuss the real-time performance of the aerial image feature matching algorithm, experiment 1 is carried out in this paper. Experiment 1 uses scale invariant feature transform (SIFT) algorithm, speeded up robust features (SURT) algorithm, rotated binary robust independent elementary features (ORB) algorithm, features from accelerated segment test (FAST) algorithm, binary robust invariant scalable key point (BRISK) algorithm, accelerated binary robust invariant scalable key point (ABRISK) algorithm, geographic information feature matching algorithm and our algorithm. According to the number of extracted feature points, the running time changes as shown in Fig 9.

Fig 9 shows that using SIFT algorithm, SURT algorithm, ORB algorithm and BRISK algorithm to extract feature points can't meet the real-time needs in aerial images with an area of

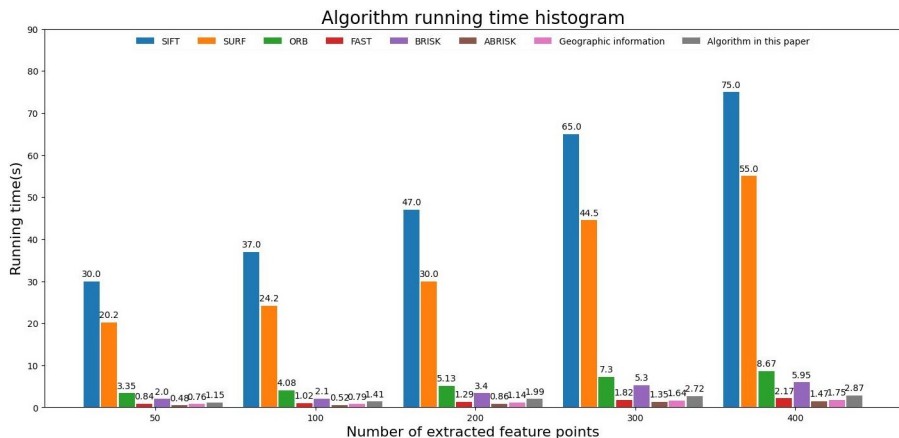

**Fig 9. Histogram of algorithm run time change.**

4864 * 3232 got by XX tilt camera. FAST algorithm, ABRISK algorithm and geographic information algorithm can meet the real-time needs, and their running time changes little with the number of feature points extracted. Because of the use of a variety of constraints to modify the extraction position of feature points, the running time of algorithm in this paper is slightly longer than that of ABRISK algorithm, but after FPGA acceleration, the running time of this algorithm meets the real-time needs of aerial camera applications. Geographic information is used for correction, which ignores most image feature information. When the swing-sweep type aerial camera gets the aerial remote sensing image, there is local geometric distortion in the image because of the change of camera swinging angle. When the sharpness of the image is gradually blur, the success rate of feature matching decreases gradually, and the change trend of the success rate is shown in Fig 10.

It is not difficult to know from Fig 10. Take BRIEF algorithm as an example to verify the geographic information correction algorithm can improve the success rate of matching. When the image is clear enough, it can be improved by 15% and 25%, while the running time is almost unchanged, as shown in Fig 10. However, the image is gradually blur, and the aerial image captured by the swing-sweep type aerial camera has local distortion, so the success rate of matching will be greatly reduced. Therefore, in the swing-sweep type aerial camera to get different sharpness of aerial images, using this algorithm for feature point matching, can improve the success rate of matching, and the running time is better than the classical feature matching algorithm, slightly longer than the geographic information correction algorithm. Considering the running time and matching precision, the algorithm can well meet the needs of aerial cameras.

To verify the feature point extraction of classical algorithms in blur aerial image, experiment 2 is designed. In the experiment, the blur aerial remote sensing image got by XX tilt measurement camera is extracted, the clear image and the blur original image are shown in Fig 11, and the extraction results of each algorithm are shown in Figs 12 and 13.

From the effect images of feature points extracted by various algorithms in blur aerial images, we can see that SURF algorithm has the largest number of feature points, and it is robust to image blur, but its extraction speed is slow, so it is difficult to meet the real-time needs. FAST algorithm and ORB algorithm are affected by image blur, the number of feature

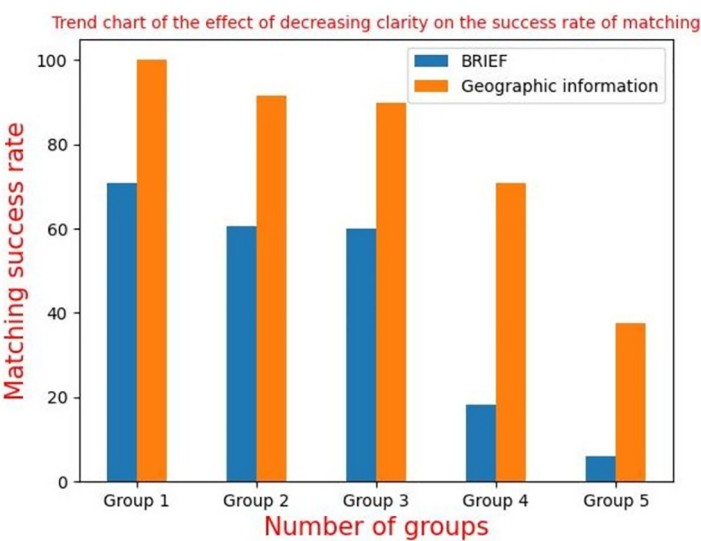

**Fig 10. Trend chart of matching success rate of sharpness reduction algorithm.**

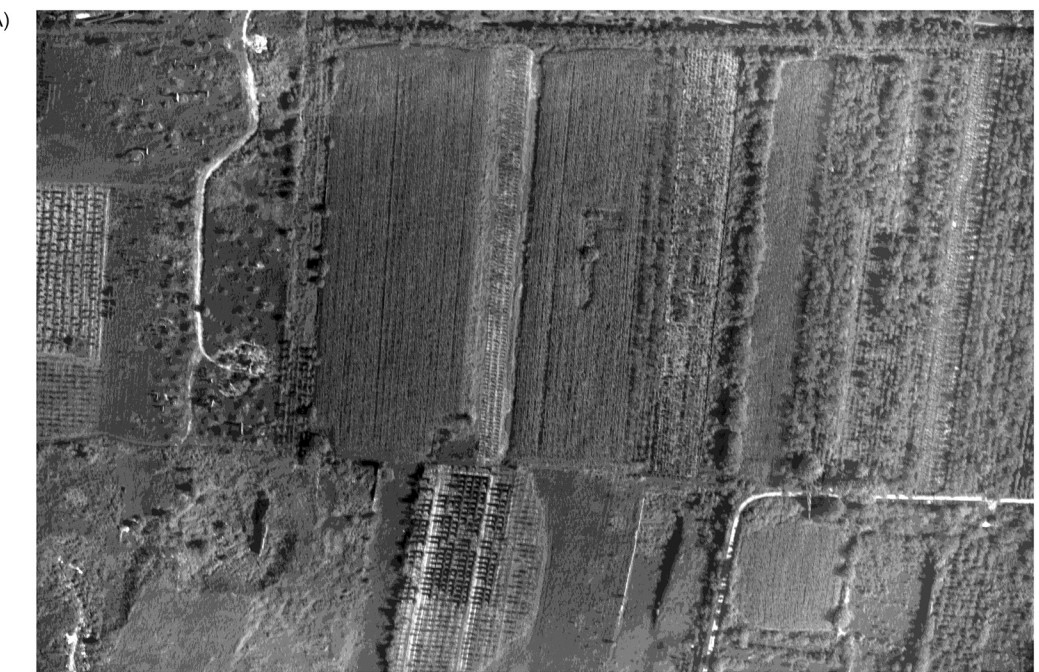

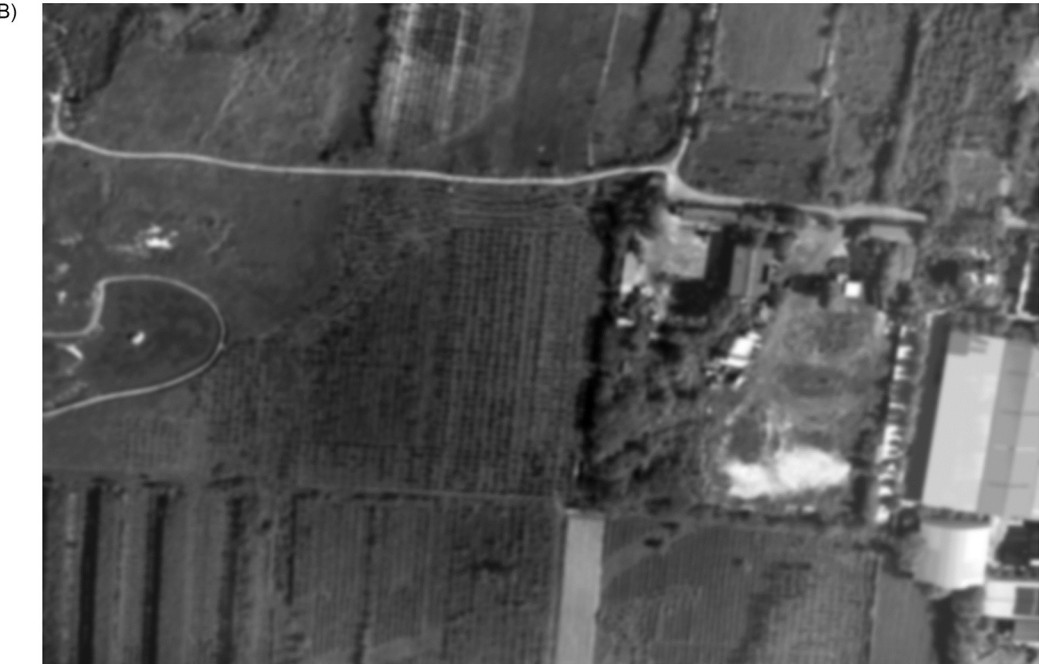

**Fig 11. The clear and the blur original aerial remote sensing images.**

(A)

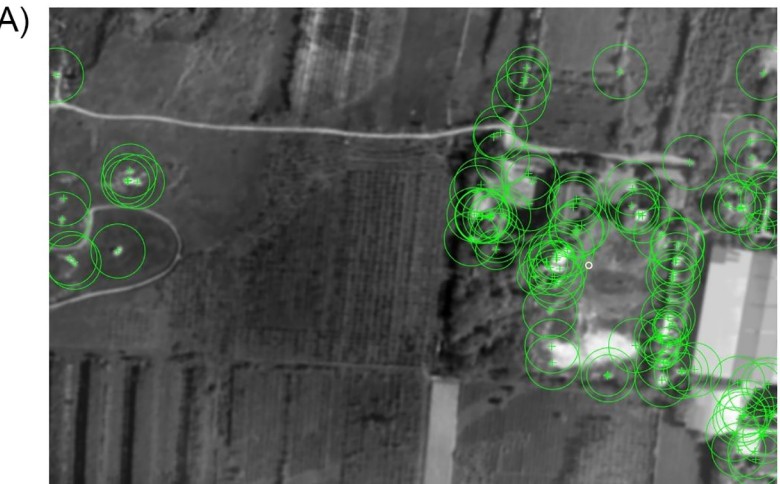

(B)

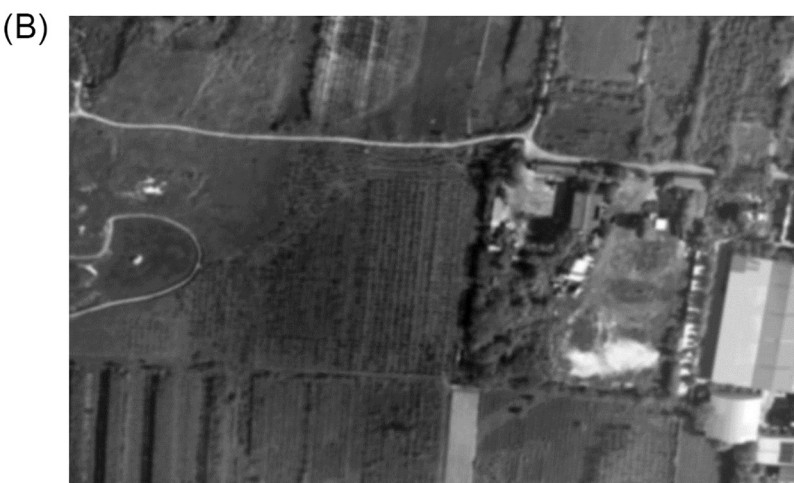

**Fig 12. The extraction results of BRISK and FAST algorithm.**

points extraction is sharply reduced, and even the possibility of extraction failure may occur in individual images. BRISK algorithm has a certain robustness to image blur, but its extraction speed still can't meet the real-time needs, so this paper uses the ABRISK algorithm which has the same extraction effect but faster extraction speed as the feature point extraction algorithm.

To verify the matching effect of this algorithm in aerial images with inconsistent sharpness, experiment 3 is designed. This experiment uses the clear aerial image of Fig 12 and the blur aerial image of Fig 13, and uses the classical feature matching algorithm and the algorithm in this paper. Fig 11 is aerial tilt images got by XX tilt camera, in which the swinging angle range of tilt aerial camera is $18°\sim55°$. High-accuracy DEM images are obtained by using InSAR tilt photography point cloud data. When the sharpness of the image is consistent, the pixel moving speed is calculated by using the extended L-k optical flow, and the pixel moving speed is calculated by using multiple swinging stripe images. The basic speed parameter is $u = 13.025 pixel/ms$, $v = 0.775 pixel/ms$ velocity, and the pixel movement speed change compensation curve between every two aerial images is shown in Figs 14 and 15.

(A)

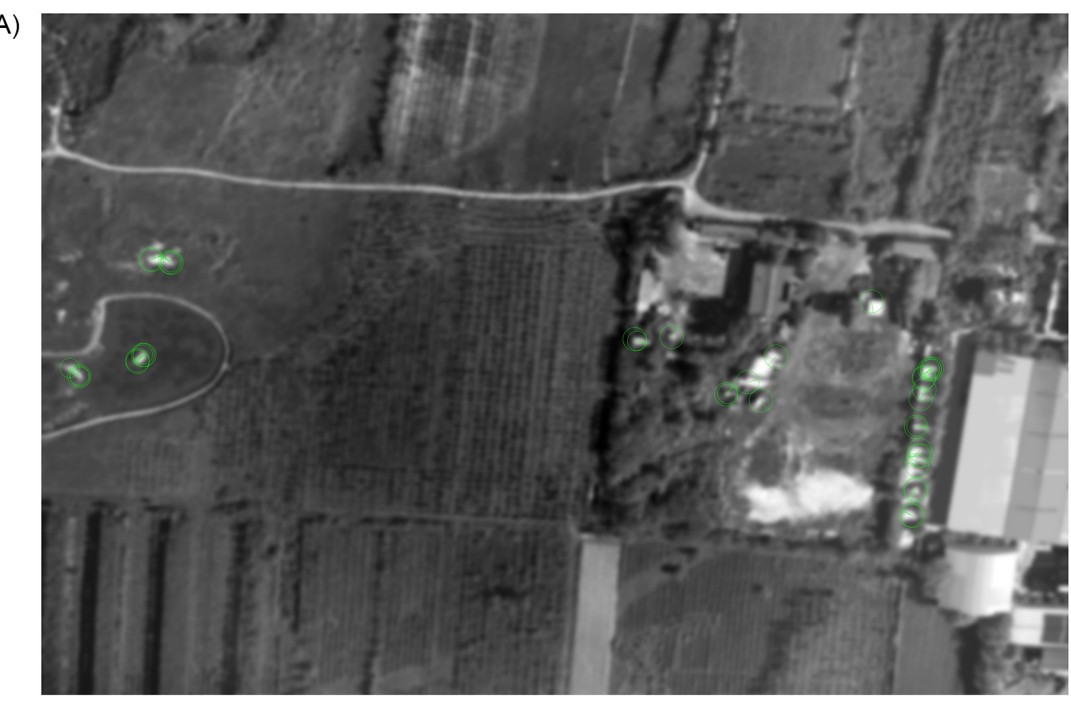

(B)

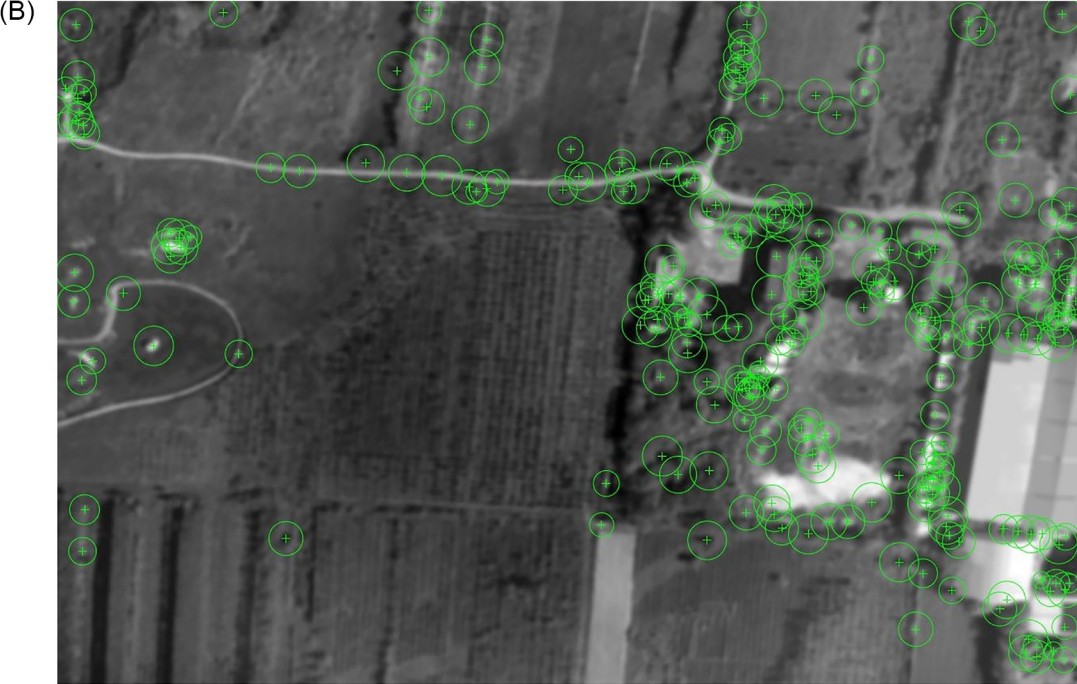

**Fig 13. The extraction results of ORB and SURF algorithm.**

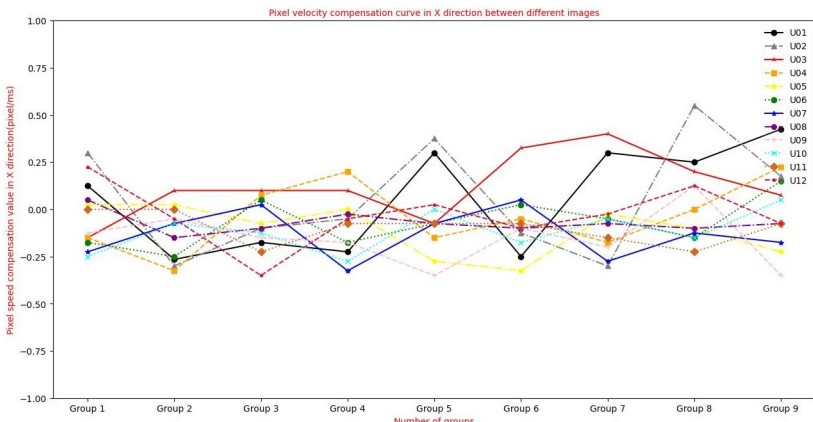

**Fig 14. Variation curve of pixel speed compensation in X direction calculated by extended L-K optical flow.**

When the sharpness of the image is consistent, the nonlinear soft margin support vector machine is used to calculate the error correction of the imaging model and record it, in which the feature point label is mainly generated automatically, followed by manual marking.

The classical feature matching algorithm and this algorithm are used to register Fig 11, in which the pixel relative compensation speed of the algorithm of this paper uses the value in Tables 1 and 2, and its effect is shown in Figs 16–19.

As can be seen from Fig 16, BRISK algorithm can still extract a considerable number of feature points in blur aerial images, but its descriptor calculation is affected by the image blur, which greatly reduces the matching precision. In the registration of multiple groups of images, it is found the precision of BRISK algorithm for registration between aerial images with inconsistent sharpness is unstable, which is greatly affected by the photographing area and blur. In addition, the FAST algorithm itself is difficult to extract feature points in blur aerial remote sensing images, so the extracted feature points are used to calculate descriptors for inconsistent aerial image registration, and there are often no correct matching point pairs. In Fig 17, the ORB algorithm is usually described by BRIEF descriptors, but because the image is blur, the gray level of the image pixel changes, which is usually different from that in the clear image, so most of the mismatched point pairs are obtained. In Fig 18, SURF algorithm extracts many

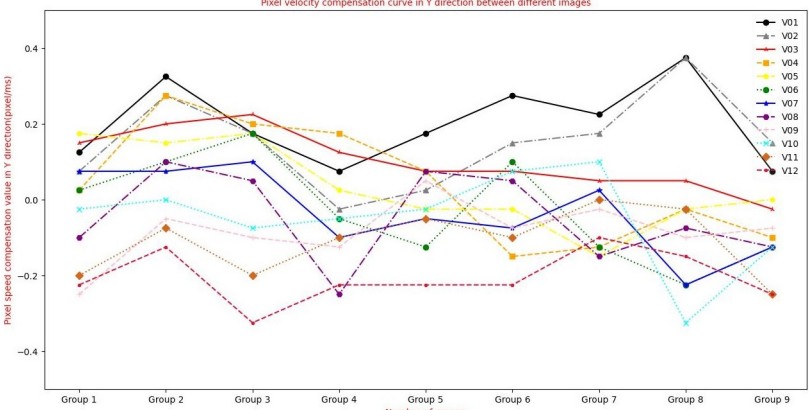

**Fig 15. Variation curve of pixel speed compensation in Y direction calculated by extended L-K optical flow.**

**Table 1. Compensation amount of pixel moving speed in X direction between different images in swinging strip.**

|        | u      | $u_1$  | $u_2$  | $u_3$  | $u_4$   | $u_5$   | $u_6$   |
|--------|--------|--------|--------|--------|---------|---------|---------|
| *value* | 13.025 | 0.100  | 0.060  | 0.120  | -0.040  | -0.110  | -0.070  |
|        | u      | $u_7$  | $u_8$  | $u_9$  | $u_{10}$ | $u_{11}$ | $u_{12}$ |
| *value* | 13.025 | -0.130 | -0.070 | -0.150 | -0.115  | -0.100  | -0.175  |

feature points from blur aerial images, but its matching time is longer, and because SURF algorithm uses Haar wavelet to get its descriptors, it is also affected by the gray change of image pixels, so there are most mismatches. Fig 19 shows the effect of feature matching using the algorithm in this paper. To ensure the position of the feature points is as uniform as possible, when utilizing ABRISK algorithm to extract the feature points, the feature points which are less than 5 pixels around the extracted position are removed. The algorithm in this paper does not rely on feature descriptor calculation, and uses geometric constraints to search, so its matching precision and computational efficiency are high. However, the modified aerial imaging model is used in this algorithm, and the local geometric distortion is too large, especially the image quality is reduced because of the large jitter or angle change of the camera in the process of getting the image, which will lead to a large matching error.

In inconsistent image sharpness, the previously got imaging model correction and the image pixel movement speed compensation in Tables 1 and 2 are used as known quantities. According to the feature points extracted from the previous image, the coordinates of the ground points are calculated by using the imaging model combined with high-accuracy DEM images, and make full use of the advantage of accurate geographic information to calculate the search range of the feature points in the later image. The algorithm determines the location of the feature points within the range determined by the relaxation factor around the image point calculated by the imaging model, and use epipolar geometric constraints to reduce the dimension. Finally, the best position is determined according to the pixel moving speed determined by the extended L-k optical flow. After the location of the feature points is finally determined, the sequential probability ratio test method is used to remove mismatched point pairs [53]. To verify the algorithm is not affected by the swinging position of the swinging strip, many matching experiments are carried out on the aerial remote sensing images got by the aerial camera when the swinging angle is large, and the matching results are shown in Fig 20.

To ensure the versatility of this algorithm, experiment 4 is designed. In this experiment, 10%, 30%, 50% Gaussian blur is added to different aerial tilt measurement images, and each feature point matching algorithm is used for feature matching. Ten groups of experiments are repeated, and the matching precision is shown in Table 3.

Experiments show the classical SURF algorithm and BRIEF algorithm can reduce the matching precision afterimage blurring, and can't complete image feature matching after blurring up to 30%. However, geographic information aided feature matching algorithm and the algorithm designed in this paper still keep the basic success rate of feature matching when the image blurring degree is less than 50%. However, geographic information aided feature

**Table 2. Compensation amount of pixel moving speed in Y direction between different images in swinging strip.**

|        | v      | $v_1$  | $v_2$  | $v_3$  | $v_4$   | $v_5$   | $v_6$   |
|--------|--------|--------|--------|--------|---------|---------|---------|
| *value* | 0.775  | 0.200  | 0.150  | 0.100  | 0.040   | 0.030   | -0.030  |
|        | v      | $v_7$  | $v_8$  | $v_9$  | $v_{10}$ | $v_{11}$ | $v_{12}$ |
| *value* | 0.775  | -0.030 | -0.050 | -0.080 | -0.050  | -0.110  | -0.205  |

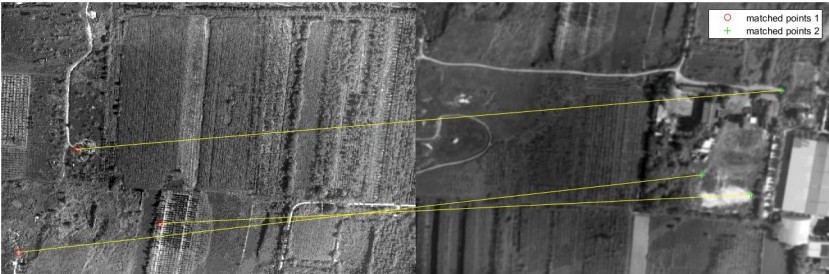

**Fig 16. Matching results of aerial images with different sharpness by BRISK algorithm.**

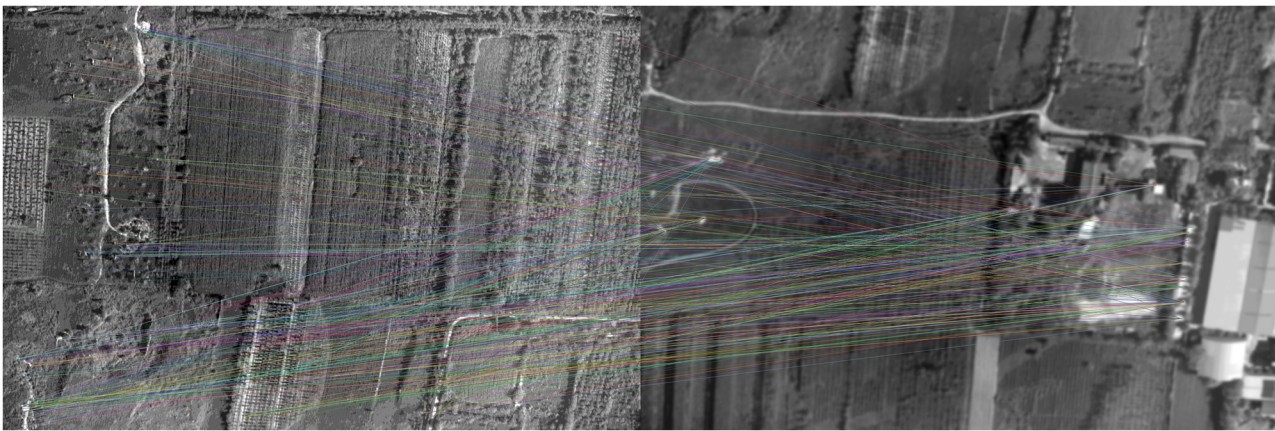

**Fig 17. Matching results of aerial images with different sharpness by BRIEF algorithm.**

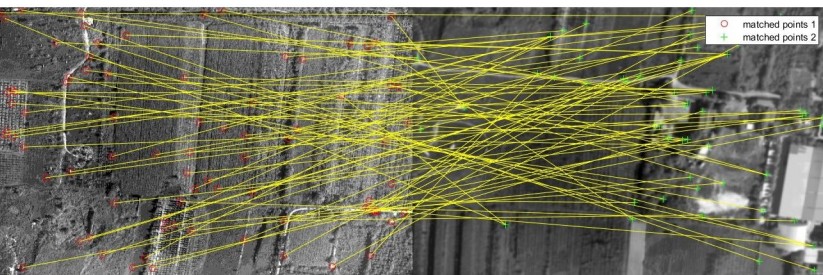

**Fig 18. Matching results of aerial images with different sharpness by SURF algorithm.**

matching algorithm can't correct the registration error caused by the imaging model, and it is affected by the local geometric distortion in the image got by the swing-sweep type aerial camera. So in the swing-sweep type aerial camera swinging process, with the increase of swinging angle and the comprehensive influence of image blur, the image registration rate will be lowered. In the algorithm of this paper, the modified aerial imaging model is used to determine the preliminary search position, and the geometric constraint and extended L-k optical flow are used to determine the location of feature points, which is not affected by the blur degree of

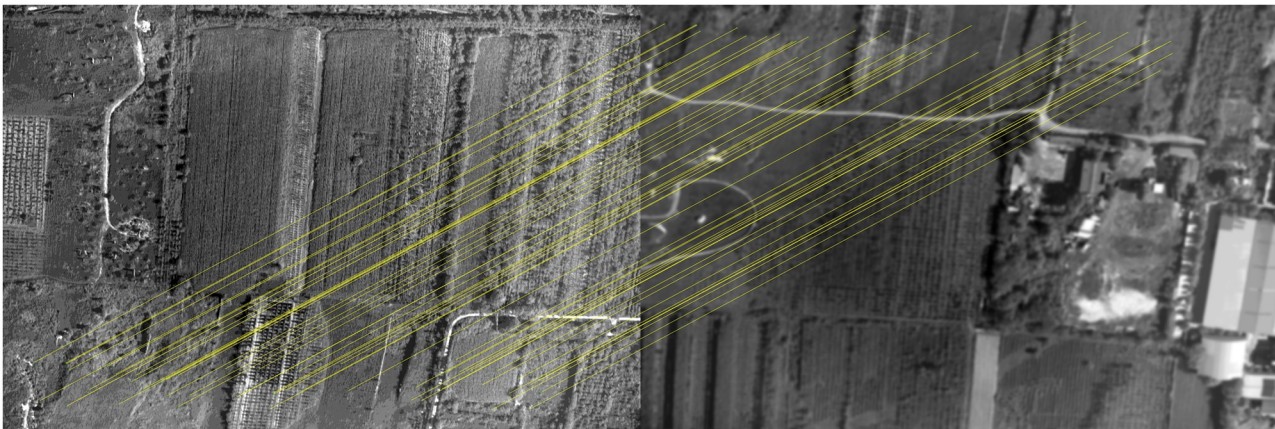

**Fig 19. Matching results of aerial images with different sharpness by algorithm in this paper.**

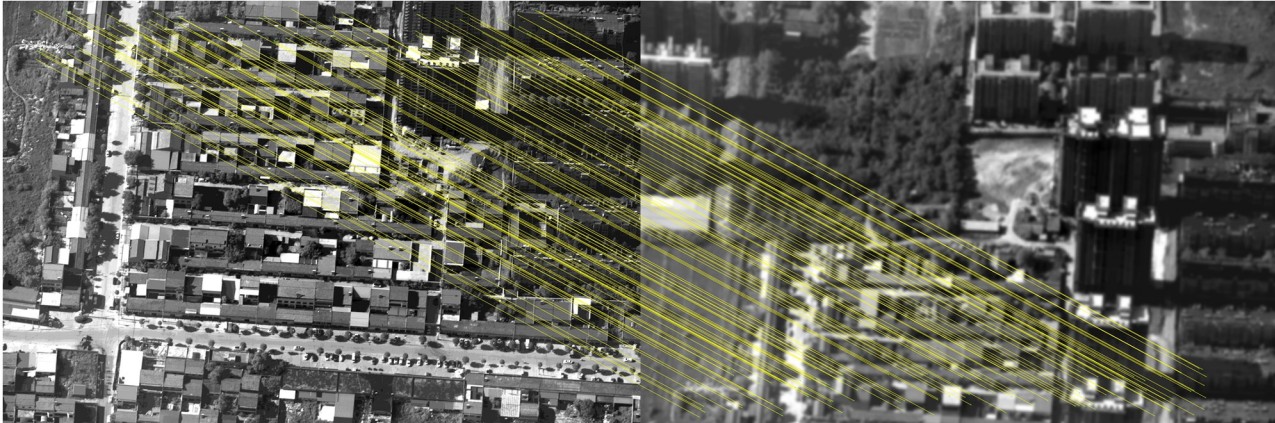

**Fig 20. Matching results of this algorithm for aerial images with different sharpness when the swinging angle is large.**

the image, and because the modified aerial imaging model and high-accuracy DEM image data are introduced, it overcomes the influence of air pressure, temperature and ground elevation difference to some extent, and further improves the success rate of registration. However, pixel movement speed compensation in the X direction calculated by this algorithm includes the influence of the forward flight speed of the aircraft, and the effect is better when the flight speed of the aircraft is almost constant. However, in sudden change of flight speed, the number of feature points will be decreased, which will lead to the failure of registration.

In this paper, high-accuracy DEM images are introduced to eliminate the influence of ground elevation difference. To verify the impact of DEM image accuracy on this algorithm, experiment 5 is designed. The experiment uses millimeter-level DEM images, centimeter-level DEM images, decimeter-level DEM images and 30-meter DEM images got by InSAR to repeat 40 experiments, and the root mean square error is calculated accordingly. The results are shown in Fig 21.

As can be observed in Fig 21, the algorithm in this paper has certain needs for the accuracy of DEM images. To ensure the success rate of feature matching, millimeter DEM images and

**Table 3. Matching precision of each feature matching algorithm in different image sharpness.**

| Image blur degree | Clear | 10% blur | 30% blur | 50% blur |
|---|---|---|---|---|
| *BRIEF* | 70% | 34% | × | × |
| *SURF* | 84% | 66% | × | × |
| *Geographic* | 100% | 88% | 70% | 40% |
| *this paper's algorithm* | 94% | 92% | 90% | 90% |

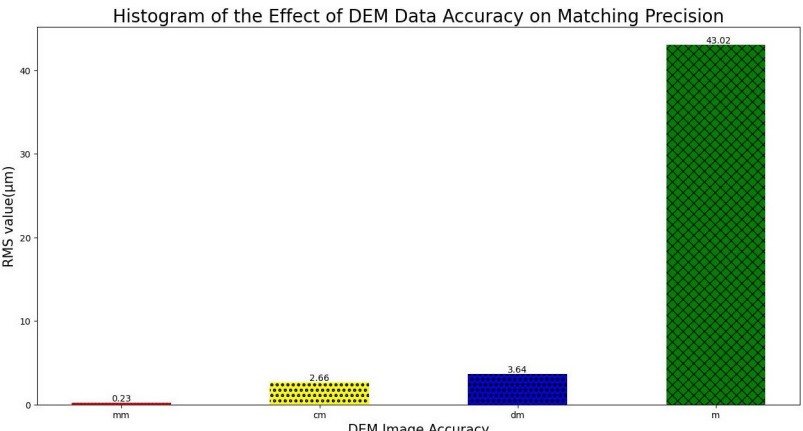

**Fig 21. Histogram of root mean square error of DEM image accuracy versus matching result.**

their data are utilized to calculate. In the figure, the influence of meter-scale DEM image data on elevation difference is small, and there is migration of geographic information, so its root mean square error is up to 43.02, which is difficult to meet the need of this algorithm.

## Discussion

Combined with the above experiments, when the number of features is 100, it takes about 1.5s to extract feature points and match them in 4864 * 3232 aerial tilt images. To ensure the real-time needs of this algorithm, hardware acceleration can be utilized to meet the real-time needs. The experimental results show the comprehensive feature matching precision of this algorithm for blur aerial images is greater than 90%.

## Conclusions

Because of air pressure, temperature, ground elevation difference and other reasons, aerial remote sensing images will be blur. When the image is blur, the matching success rate of the classical feature matching algorithm decreases rapidly, and the image sharpness detection result can't be obtained according to the feature point matching. Therefore, an algorithm is needed to maintain a high feature matching rate when aerial images are blur to meet the needs. In this paper, according to the characteristics of overlapping area between the two images pervious and latter photographing by swing-sweep type aerial camera, the algorithm is designed by using geographic information and high-accuracy DEM image. To solve the problem of feature matching between aerial images with different sharpness, an aerial image feature matching algorithm based on multi-constraints is proposed. That is, on the premise of getting high-accuracy DEM images and photographing a series of clear images, using a variety of

classical computer vision algorithms, such as nonlinear soft margin support vector machine, extended L-k optical flow, polar constraint and so on, in modifying the imaging model and obtaining the relative compensation of pixel moving speed, the feature matching between aerial images with different sharpness is realized. It can achieve 90% of the feature matching precision in the case of blur, and the speed can reach 1.5s when the number of features is 100. After hardware acceleration such as FPGA, it can fully meet the real-time needs of aerial cameras, which provides a necessary prerequisite for aerial cameras to use feature point matching to get image sharpness detection results.

## Supporting information

**S1 Data.**
(ZIP)

## Author Contributions

**Data curation:** Dongchen Dai, Yu Zhang, Haijiang Wang.

**Methodology:** Dongchen Dai.

**Resources:** Lina Zheng, Guoqin Yuan, He Zhang.

**Software:** Dongchen Dai, Guoqin Yuan, He Zhang, Qi Kang.

**Supervision:** Lina Zheng, Guoqin Yuan.

**Validation:** Lina Zheng, Guoqin Yuan, Haijiang Wang.

**Visualization:** Dongchen Dai, Yu Zhang, Qi Kang.

**Writing – original draft:** Dongchen Dai.

**Writing – review & editing:** Dongchen Dai.

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
