## [Decision Letter · Decision Letter 0]

18 Jul 2022

PONE-D-22-16720Real-time and High Precision Feature Matching Between Blur Aerial ImagesPLOS ONE

Dear Dr. Dai,

Thank you for submitting your manuscript to PLOS ONE. After careful consideration, we feel that it has merit but does not fully meet PLOS ONE’s publication criteria as it currently stands. Therefore, we invite you to submit a revised version of the manuscript that addresses the points raised during the review process.

We look forward to receiving your revised manuscript.

Kind regards,

Xuejian Wu, Ph.D.

Academic Editor

PLOS ONE

Journal Requirements:

2. Please ensure that you refer to Figures 3, 4 and 9 in your text as, if accepted, production will need this reference to link the reader to the figure.

Reviewers' comments:

Reviewer's Responses to Questions

**Comments to the Author**

1. Is the manuscript technically sound, and do the data support the conclusions?

Reviewer #1: Yes

Reviewer #2: Partly

2. Has the statistical analysis been performed appropriately and rigorously? 

Reviewer #1: Yes

Reviewer #2: N/A

3. Have the authors made all data underlying the findings in their manuscript fully available?

Reviewer #1: Yes

Reviewer #2: Yes

4. Is the manuscript presented in an intelligible fashion and written in standard English?

Reviewer #1: Yes

Reviewer #2: No

5. Review Comments to the Author

Reviewer #1: The paper designed a new algorithm to match features between blur aerial Images. With the constraints of DEM, polar geometry and optical flow, good results were obtained from the new algorithm.

Recommend to show more images with matching results.

Reviewer #2: This paper presents a complex approach to achieve real-time and high precision feature matching between blur aerial images. However, great efforts are required to improve the quality of this paper. Major concerns are listed below for the authors’ reference.

1. Proofreading is necessary to correct language and grammar errors and incomprehensible presentations. Terminologies and acronyms (the full names must be presented the first time they appear) should be consistent and well defined throughout the whole paper. Descriptions and explanations of the figures are necessary instead of only showing them in the paper, and the quality of figures can be improved, e.g., fig. 5 provides too complicated information, please simplify it.

2. Please illustrate the objective of matching blur images in terms of its applicability. In practice, blur images are avoided during the mission plan stage in a standard photogrammetry mission as they cannot produce authentic products. In computer vision, blur images are also filtered out because they increase the difficulty of data processing. Thus, I am curious about under what conditions/situations shall this method yield its capacity.

3. The literature review in introduction is outdated and inadequate. More advanced local feature matching methods, such as AB-SIFT, BRISK, LATCH, ABRISK, SC-EABRISK, BEBLID, have to be reviewed and discussed as well. I strongly suggest the authors to select one of the approaches to replace the ORB method. Please refer to the following links:

https://ieeexplore.ieee.org/document/7095554

https://ieeexplore.ieee.org/document/6126542

https://doi.org/10.1016/j.isprsjprs.2017.03.017

https://doi.org/10.1016/j.patrec.2020.04.005

https://ieeexplore.ieee.org/document/8766118

https://www.mdpi.com/1424-8220/21/18/6035

4. In p.2, the authors mixed and misused two terms in terms of matching accuracy and matching precision. Accuracy assessment requires ground truth data, while precision evaluation dose not. Similar issue is also for ‘precise’ DEM. Please check the following literatures to clarify their definitions.

https://www.mdpi.com/2072-4292/10/5/747

https://ieeexplore.ieee.org/document/8859298

5. In Eq. 2, please describe why the correction is added to the roll angle, and what is its relationship among the axes shown in Fig. 3 because the definition of a swing sweep camera is not available.

6. It is unclear why the soft-margin SVM is used to address the issue. Please provide relevant literatures or your considerations/explanations to support the utilization of this method. Also, there is a sudden jump from soft-margin SVM to aerial imaging model without any logic connection. The authors can consider to reorganize these paragraphs in a logical order to express the purpose of this section. The definition of the constant ‘C’ in Eq. 4 is missing.

7. In p.7, the epipolar constraint is to reduce the feature matching dimension (i.e. from 2D to 1D search) rather than to extract feature points. Thereafter, area-based feature is defined/extracted, and then searching for their correspondence is performed on the epipolar line, which is a common approach in photogrammetry and computer vision. As the authors exploit ORB features for matching, how to guarantee those features are right on the epipolar lines is problematic, and thus the solution might be by chance and unstable.

8. The ORB itself is a feature extraction algorithm but not a matching algorithm, and it uses Hamming distance to measure the feature similarity. Please carefully check the literatures shown in 4 and 5 to understand the differences between feature extraction and feature matching as there are misused terms in the paper.

9. Please explain the resolution of the DEM and discuss whether the matching result is affected by the DEM’s resolution.

10. More experimental results are needed to discuss and support that their proposed method is effective.

6. PLOS authors have the option to publish the peer review history of their article (what does this mean?). If published, this will include your full peer review and any attached files.

Reviewer #1: No

Reviewer #2: No

---

## [Author Response · Author response to Decision Letter 0]

24 Aug 2022

Response letter

Dear editors of "PLoS ONE" and reviewers:

Hello!

Thank you very much for reviewing the article "Real-time and High Precision Feature Matching Between Blur Aerial Images" (Manuscript Number: PONE-D-22-16720). I have made a detailed answer to the questions raised by the reviewers, and made detailed amendments to the paper according to the opinions of the reviewers. The original text is covered with deletion lines, and the revised part is highlighted in yellow.

Thank you very much for the reviewers' comments on this article. The questions you asked about this article are precisely the omissions in the writing of this article. The following will respond to your questions one by one:

Manuscript reviewer 1:

（1） The paper designed a new algorithm to match features between blur aerial Images. With the constraints of DEM, polar geometry and optical flow, good results were obtained from the new algorithm. Recommend to show more images with matching results.

Answer: Thank you very much for the approval of the method of the paper by the reviewers. I made reference to your opinion and added a comparison diagram of the matching effect of different algorithms in the experimental part of the paper. In order to verify the effectiveness of this algorithm at the swinging angle of the pendulum, experiment 3 is added. In order to verify the feature matching accuracy of this algorithm, experiment 4 is designed. In experiment 4, the proposed method is used to match the images in different swinging stripes, and the effect is compared with the classical feature matching algorithm. After repeated experiments, the relevant data are obtained, and the results are shown in Table 2. The revised part has been marked in the revised manuscript, please download and check it.

Manuscript reviewer 2:

（1） Proofreading is necessary to correct language and grammar errors and incomprehensible presentations. Terminologies and acronyms (the full names must be presented the first time they appear) should be consistent and well defined throughout the whole paper. Descriptions and explanations of the figures are necessary instead of only showing them in the paper, and the quality of figures can be improved, e.g., fig. 5 provides too complicated information, please simplify it.

Answer: Thank you very much for your comments. I am very sorry that this article failed to find language and grammatical errors in time before the initial submission. In this revision, I checked the language and grammar in the paper many times, corrected the errors in the first draft one by one, and examined the added parts in detail. In this revision, I added the full name to the terms and the location of the acronym that appear in the paper, and kept the meaning of the article consistent. The images in the article are simplified and necessary explanations are added. The specific changes have been marked in the revised draft, please download and review.

（2） Please illustrate the objective of matching blur images in terms of its applicability. In practice, blur images are avoided during the mission plan stage in a standard photogrammetry mission as they cannot produce authentic products. In computer vision, blur images are also filtered out because they increase the difficulty of data processing. Thus, I am curious about under what conditions/situations shall this method yield its capacity.

Answer: Thank you very much for your attention to the method of this article. The use of blurred images is not used in standard photogrammetric tasks and computer vision, but in real-time detection of image sharpness. After the feature matching is realized between the images with different sharpness, according to the offset position of the feature matching points and applying the optical system principles such as Gaussian imaging principle, the image clarity detection parameters can be obtained and used as the basic parameters to adjust the image to get a sharp image. In the process of revising the manuscript, in order to ensure the integrity of the paper, the application field will be described in the introduction of the article. The revised part has been marked in the revised draft, please download and view it.

（3） The literature review in introduction is outdated and inadequate. More advanced local feature matching methods, such as AB-SIFT, BRISK, LATCH, ABRISK, SC-EABRISK, BEBLID, have to be reviewed and discussed as well. I strongly suggest the authors to select one of the approaches to replace the ORB method. Please refer to the following links:

https://ieeexplore.ieee.org/document/7095554

https://ieeexplore.ieee.org/document/6126542

https://doi.org/10.1016/j.isprsjprs.2017.03.017

https://doi.org/10.1016/j.patrec.2020.04.005

https://ieeexplore.ieee.org/document/8766118

https://www.mdpi.com/1424-8220/21/18/6035

Answer: Thank you very much for your comments. In this revision, the introduction will be rewritten, and with reference to a number of literature reviews in the field, the methods are partially summarized and described. The revised part has been marked in the revised draft, please review it. In this paper, when selecting ORB algorithm as the feature extraction algorithm in this paper, we mainly consider that the running time of the algorithm needs to meet the real-time requirements, and fail to comprehensively consider many factors. Thank you very much for the several references provided by the manuscript review teacher, which opened up new ideas for me. In the process of revising this manuscript, I use ABRISK algorithm instead of ORB algorithm for feature extraction, the running time of this algorithm is not sensitive to image size, and the extraction speed is faster, so it is very suitable for this method. ORB algorithm is introduced into the experiment as one of the comparison methods. This method does not choose the faster EABRISK algorithm, which takes into account the situation of this method in the application field. Sharpness detection is usually calculated and obtained according to the photography sequence, but the Interactive Two-Side Matching (ITSM) theory of EABRISK algorithm is difficult to meet in the application field of this method, so this paper uses ABRISK method for feature extraction.

（4） In p.2, the authors mixed and misused two terms in terms of matching accuracy and matching precision. Accuracy assessment requires ground truth data, while precision evaluation dose not. Similar issue is also for ‘precise’ DEM. Please check the following literatures to clarify their definitions.

https://www.mdpi.com/2072-4292/10/5/747

https://ieeexplore.ieee.org/document/8859298

Answer: Thank you very much for the problems pointed out by the reviewer. In my usual use, I do not distinguish between "precision" and "accuracy". After reviewers reminder, in the future use, will pay attention to the difference between the "precision" and "accuracy", and use in the correct situation. In this revision, the use of "precision" and "accuracy" in the article has been all revised, and the additions have also been reviewed accordingly. The revised part has been marked in the revised draft, please download and view it.

（5） In Eq. 2, please describe why the correction is added to the roll angle, and what is its relationship among the axes shown in Fig. 3 because the definition of a swing sweep camera is not available.

Answer: Thank you very much for your comments. I am very sorry that the definition of the swing-sweep type aerial camera was not described at the time of the initial submission, which led to the unclear description of formula 2 and figure 3. In this revision, I describe the classification and working mode of aerial cameras, and the characteristics of obtaining aerial remote sensing images in the introduction, and describe formula 2 and figure 3 accordingly. This part of the revision has been marked in the revised draft, please download and view it.

（6） It is unclear why the soft-margin SVM is used to address the issue. Please provide relevant literatures or your considerations/explanations to support the utilization of this method. Also, there is a sudden jump from soft-margin SVM to aerial imaging model without any logic connection. The authors can consider to reorganize these paragraphs in a logical order to express the purpose of this section. The definition of the constant ‘C’ in Eq. 4 is missing.

Answer: Thank you very much for your comments. In image processing, support vector machine has been widely used in feature classification, image segmentation, remote sensing image processing and other fields. In the experiment, I found that a certain amount of pixel offset will be produced when the modified aerial imaging model is used for feature matching. The offset is related to the swinging position of the swing-sweep type aerial camera when obtaining aerial remote sensing images. if the KNN algorithm is used for matching, it will result in a mismatch of about 27.43%. Therefore, this paper uses nonlinear soft margin support vector machine to calculate different relaxation factors according to the different swinging position of the swing-sweep type aerial camera. According to the relaxation factor and the corresponding point position determined by the aerial imaging model, the search range of feature points is reduced as much as possible to improve the efficiency of feature matching. In addition, this paper refers to the method in the literature when using nonlinear soft-interval support vector machine because of the asymmetry of data. In order to ensure the integrity of this article, the revised content is reflected in the introduction of the revised draft, and the corresponding part is readjusted and supplemented. In this revision, the lack of definition in Formula 4 has been supplemented, and other formulas have been reviewed accordingly, and there is no similar situation. The specific changes are marked in the revised draft, please download and view them. The relevant contents of this section quote the following references:

https://ieeexplore.ieee.org/abstract/document/7882747

https://www.sciencedirect.com/science/article/abs/pii/S092427160700055X

https://ieeexplore.ieee.org/abstract/document/1641014/

https://link.springer.com/chapter/10.1007/11941439_30

（7） In p.7, the epipolar constraint is to reduce the feature matching dimension (i.e. from 2D to 1D search) rather than to extract feature points. Thereafter, area-based feature is defined/extracted, and then searching for their correspondence is performed on the epipolar line, which is a common approach in photogrammetry and computer vision. As the authors exploit ORB features for matching, how to guarantee those features are right on the epipolar lines is problematic, and thus the solution might be by chance and unstable.

Answer: Thank you very much for your comments. In this revision, the relevant description is modified according to the opinions of the reviewer. ABRISK is used to replace ORB algorithm, and the corresponding problems are redescribed in the section of epipolar geometric constraints. This part of the content has been marked in the revision, please review the teacher to download and view. 

（8） The ORB itself is a feature extraction algorithm but not a matching algorithm, and it uses Hamming distance to measure the feature similarity. Please carefully check the literatures shown in 4 and 5 to understand the differences between feature extraction and feature matching as there are misused terms in the paper.

Answer: Thank you very much for your comments. I am very sorry that there was a mistake in the description of this part due to carelessness in the process of writing the first draft. In this revision, I modified the corresponding terms, the added content has also been reviewed, there are no similar problems. The revised part has been marked in the revised manuscript, please download and check it. 

（9） Please explain the resolution of the DEM and discuss whether the matching result is affected by the DEM’s resolution.

Answer: Thank you very much for your comments. When writing the first draft, I did not discuss this part. After being reminded by the teacher to review the manuscript, experiment 4 is added to the experiment part. The millimeter, centimeter, decimeter and 30m DEM images obtained by InSAR are applied to this algorithm, and the calculation is repeated 40 times to obtain the root mean square error. The corresponding results are reflected in figure 25. This part of the revision has been marked in the revised draft, please download and view it.

（10） More experimental results are needed to discuss and support that their proposed method is effective.

Answer: Thank you very much for your comments. In the process of revising the manuscript, I refer to the references provided by you to expand the content of the experiment. In experiment 1, the running speed of this algorithm is compared with that of classical algorithm, and the real-time performance of this algorithm is verified. Experiment 2 compares the feature extraction effect of classical algorithms on blurred images, and describes the reason why ABRISK is selected as the feature extraction algorithm. Experiment 3 compares the feature matching effect of the classical algorithm and the proposed algorithm on aerial images with different sharpness, and proves that this algorithm can realize the feature matching of aerial images with different sharpness. Experiment 4 verifies the wide applicability of this algorithm, matches the features of aerial remote sensing images in different swinging stripes, and calculates the success rate of feature matching. The specific results are reflected in Table 2. Experiment 5 uses DEM images with different accuracy, and discusses the influence of citing DEM image accuracy on the precision of the algorithm. This part of the content has been marked in the revised draft, please download and view it.

I am very grateful to the “PLoS ONE” editor teacher and the manuscript reviewer for correcting the errors in this article. I have revised the questions raised by the manuscript reviewer and marked them in the corresponding position of the revised manuscript. 

The above is my answer to the questions raised by the experts. Please review by the editors and reviewers.

 Author: Dongchen Dai

 School: Changchun Institute of Optics and Fine Machinery and Physics, Chinese Academy of Sciences

 Date: August 24, 2022

---

## [Editor Report · Decision Letter 1]

4 Sep 2022

Real-time and High Precision Feature Matching Between Blur Aerial Images

PONE-D-22-16720R1

Dear Dr. Dai,

We’re pleased to inform you that your manuscript has been judged scientifically suitable for publication and will be formally accepted for publication once it meets all outstanding technical requirements.

Kind regards,

Xuejian Wu, Ph.D.

Academic Editor

PLOS ONE
---

## [Editor Report · Acceptance letter]

8 Sep 2022

PONE-D-22-16720R1 

Real-time and High Precision Feature Matching Between Blur Aerial Images 

Dear Dr. Zheng:

I'm pleased to inform you that your manuscript has been deemed suitable for publication in PLOS ONE. Congratulations! Your manuscript is now with our production department. 

Kind regards, 

on behalf of

Dr. Xuejian Wu 

Academic Editor

PLOS ONE